# Epimutations are associated with CHROMOMETHYLASE 3-induced de novo DNA methylation

Jered M Wendte[1†], Yinwen Zhang[2†], Lexiang Ji[2†], Xiuling Shi[1], Rashmi R Hazarika[3], Yadollah Shahryary[3], Frank Johannes[3,4], Robert J Schmitz[1]*

[1]Department of Genetics, University of Georgia, Athens, United States; [2]Institute of Bioinformatics, University of Georgia, Athens, United States; [3]Department of Plant Science, Technical University of Munich, Freising, Germany; [4]Institute for Advanced Study, Technical University of Munich, Garching, Germany

**Abstract** In many plant species, a subset of transcribed genes are characterized by strictly CG-context DNA methylation, referred to as gene body methylation (gbM). The mechanisms that establish gbM are unclear, yet flowering plant species naturally without gbM lack the DNA methyltransferase, CMT3, which maintains CHG (H = A, C, or T) and not CG methylation at constitutive heterochromatin. Here, we identify the mechanistic basis for gbM establishment by expressing *CMT3* in a species naturally lacking *CMT3. CMT3* expression reconstituted gbM through a progression of de novo CHG methylation on expressed genes, followed by the accumulation of CG methylation that could be inherited even following loss of the *CMT3* transgene. Thus, gbM likely originates from the simultaneous targeting of loci by pathways that promote euchromatin and heterochromatin, which primes genes for the formation of stably inherited epimutations in the form of CG DNA methylation.

DOI: https://doi.org/10.7554/eLife.47891.001

*For correspondence:
schmitz@uga.edu

†These authors contributed equally to this work

Competing interests: The authors declare that no competing interests exist.

## Introduction

Heritable gains or losses of DNA methylation, or epimutations, can have important phenotypic consequences. Examples include peloric mutants of toadflax, where differential methylation of a single transcription factor can change flowers from bilaterally to radially symmetrical and the colorless non-ripening locus of tomato, where differential methylation affects fruit ripening (*Cubas et al., 1999*; *Manning et al., 2006*). Despite the potential implications for phenotypic change, little is known about how epimutations form.

One form of genic DNA methylation that may provide clues to the mechanisms of epimutation is gene body methylation (gbM), which is found on a subset of expressed genes in many eukaryotic genomes, including most flowering plants (*Bewick et al., 2017*; *Bewick and Schmitz, 2017*; *Cokus et al., 2008*; *Feng et al., 2010*; *Huff and Zilberman, 2014*; *Lister et al., 2008*; *Niederhuth et al., 2016*; *Regulski et al., 2013*; *Seymour et al., 2014*; *Takuno and Gaut, 2013*; *Takuno et al., 2016*; *Tran et al., 2005*; *Wang et al., 2015*; *Zemach et al., 2010*; *Zhang et al., 2006*; *Zilberman et al., 2007*). GbM has been functionally implicated in certain processes, including transcriptional regulation, transcript processing, and suppression of intragenic transcripts (*Bewick and Schmitz, 2017*; *Choi et al., 2019*; *Lorincz et al., 2004*; *Maunakea et al., 2010*; *Regulski et al., 2013*; *Vial-Pradel et al., 2018*; *Zilberman et al., 2008*; *Zilberman et al., 2007*). However, natural and experimental losses of gbM have also been documented with no obvious effects on expression or chromatin structure (*Bewick et al., 2016*; *Bewick et al., 2019*; *Niederhuth et al., 2016*; *Zhang et al., 2006*). One hypothesis that may reconcile these seemingly

conflicting results explains the formation of gbM as a passive process resulting from transient localization of the proteins that maintain heterochromatin to genic space (i.e. euchromatin) (*Bewick et al., 2016*; *Bewick and Schmitz, 2017*; *Inagaki and Kakutani, 2012*; *Teixeira and Colot, 2009*; *Wendte and Schmitz, 2018*). Here, gbM is proposed to arise passively, but, once established, is maintained due to the preferential recruitment of maintenance methyltransferases to previously methylated sites. Under this line of reasoning, gbM may have functional consequences in some cases, but can also be non-functional, which could explain experimental findings. A better understanding of the mechanistic basis for the establishment of gbM will provide important insights into its possible functions, the mechanisms underlying epimutations, as well as to factors that might influence DNA methyltransferase mistargeting in disease states (*Wendte and Schmitz, 2018*).

In plants, most knowledge of DNA methylation is derived from studies of *Arabidopsis thaliana*, where cytosines in different sequence contexts are the preferred substrates of distinct pathways. The RNA directed DNA methylation (RdDM) pathway acts to de novo methylate cytosines in all sequence contexts (*Matzke and Mosher, 2014*). In RdDM, DOMAINS REARRANGED METHYLTRANSFERASE 2 (DRM2) is targeted to chromatin via non-coding RNAs produced by two plant-specific multi-subunit RNA polymerases (*Haag and Pikaard, 2011*; *Wendte and Pikaard, 2017*; *Zhou and Law, 2015*). Following de novo establishment, methylation of symmetrical CG cytosines is maintained by METHYLTRANSFERASE 1 (MET1), which is recruited to hemi-methylated CG sites to methylate the complementary, unmethylated CG sites (*Finnegan et al., 1996*; *Ronemus et al., 1996*; *Woo et al., 2008*; *Woo et al., 2007*). Symmetrical CHG (H = A, T, or C) cytosine methylation is maintained by CHROMOMETHYLASE 3 (CMT3) (*Bartee et al., 2001*; *Lindroth et al., 2001*; *Papa et al., 2001*). Among CHG sites, CMT3 shows a marked preference for CWG (W = A or T) cytosines relative to CCG cytosines (*Gouil and Baulcombe, 2016*; *Stoddard et al., 2019*). An additional CHROMOMETHYLASE, CMT2, targets the methylation of CHH cytosines (*Stroud et al., 2014*; *Zemach et al., 2013*). In *A. thaliana*, CHH sites targeted by CMT2 can be distinguished from those targeted by RdDM, as they show an enrichment in CWA methylation relative to other CHH contexts, in contrast to RdDM target regions which show no preferred site enrichment (*Gouil and Baulcombe, 2016*).

CMT2 and CMT3 participate in a self-reinforcing feedback loop with an additional heterochromatin modification, histone H3 lysine nine di-methylation (H3K9me2) (*Du et al., 2015*; *Du et al., 2012*; *Stoddard et al., 2019*; *Stroud et al., 2014*). CMT2 and CMT3 are both thought to depend on direct physical binding to H3K9me2 for targeting to chromatin and methylation (*Du et al., 2012*; *Stoddard et al., 2019*; *Stroud et al., 2014*). DNA methylation can be physically bound by the H3K9 methyltransferases, which reinforces the co-localization of H3K9me2 and CMT-dependent DNA methylation (*Bernatavichute et al., 2008*; *Du et al., 2014*; *Du et al., 2015*; *Johnson et al., 2007*; *Li et al., 2018*). This co-dependency results in losses of CMT-dependent DNA methylation in H3K9 methyltransferase mutants, as well as losses of H3K9me2 in *cmt* mutants (*Du et al., 2015*; *Jackson et al., 2002*; *Malagnac et al., 2002*; *Mathieu et al., 2005*; *Soppe et al., 2002*; *Stroud et al., 2014*; *Stroud et al., 2013*; *Tariq et al., 2003*).

GbM is restricted to CG context cytosines (*Bewick and Schmitz, 2017*; *Cokus et al., 2008*; *Lister et al., 2008*; *Niederhuth et al., 2016*; *Takuno et al., 2016*; *Tran et al., 2005*), and therefore dependent on MET1 (*Cokus et al., 2008*; *Lister et al., 2008*). However, complementation of *met1* mutants with wild-type *MET1* fails to restore gbM, presumably because hemi-methylated CG cytosines required for MET1 recruitment have been lost (*Bewick et al., 2016*; *Reinders et al., 2009*). Thus, the mechanisms required for the establishment of gbM are unclear.

Recent comparative analyses have identified Angiosperm plant species lacking gbM (*Bewick et al., 2016*; *Niederhuth et al., 2016*). Concurrent with the loss of gbM is the loss of the gene encoding CMT3, which has led to the hypothesis that CMT3 is required for the initial establishment of gbM (*Bewick et al., 2016*; *Bewick et al., 2017*; *Bewick and Schmitz, 2017*). The lack of immediate restoration of gbM in *MET1* complemented *met1* mutants has led to the proposition that CMT3 is only rarely localized to gene bodies such that gbM only accumulates slowly over evolutionary time scales (*Bewick et al., 2016*; *Bewick and Schmitz, 2017*; *Inagaki and Kakutani, 2012*; *Wendte and Schmitz, 2018*). Direct experimental testing of this model is difficult, and it is also currently unclear whether CMT3 can be active at regions with no pre-existing DNA or H3K9 methylation. Furthermore, the favored activity of ZMET2, the maize ortholog of CMT3, is a maintenance methyltransferase of hemi-methylated cytosines in the CHG sequence context (*Stoddard et al.,*

*2019*), and, thus, a de novo activity that results in methylation of CG cytosines in vivo is unprecedented.

To gain insights into this process and provide a direct test of a role for CMT3 in gbM, we heterologously expressed *A. thaliana CMT3 (AtCMT3)* in a plant species that has lost both *CMT3* and gbM, *Eutrema salsugineum* (*Bewick et al., 2016*). *Eutrema salsugineum*, like *A. thaliana*, is a member of the Brassicaceae family and diverged from a common ancestor with *A. thaliana* ~47 million years ago (*Arias et al., 2014*). Expression of *AtCMT3* in *E. salsugineum* resulted in gains of CHG methylation over repetitive sequences characterized by the presence of H3K9me2, as predicted based on the known mechanism of CMT3 targeting. However, *AtCMT3* expressing lines also exhibited ectopic CHG methylation over a subset of genes in an *AtCMT3*-expression dependent manner. Genes that gained CHG methylation were orthologs of *A. thaliana* gbM genes and had no prior CHG methylation or detectable H3K9 methylation, suggesting de novo methylation activity of CMT3 in vivo. Unexpectedly, gains of CHG methylation did not result in stable accumulation of H3K9 methylation over gene bodies or expression changes, showing that the genic CHG methylation was uncoupled from transcriptional silencing, similar to gbM, likely due to transcription-coupled de-methylases. Gains in CHG methylation were also associated with gains in CG and CHH methylation, and removal of the transgene via crossing to non-transgenic parents, or progressive *AtCMT3* transgene silencing over six generations of propagation, revealed that ectopic genic CG methylation was preferentially maintained relative to genic CHG or CHH methylation following the loss of *AtCMT3* expression. The results provide new insights to the mechanism of CMT3-initiation of gbM by demonstrating that CMT3 promotes the establishment of genic CG epimutations which can be maintained even in the absence of *CMT3*.

## Results

### Expression of AtCMT3 in *E. salsugineum* results in increased CHG methylation

To gain insights into the mechanisms of CMT3-targeting of DNA methylation, full-length genomic *CMT3* from *A. thaliana (AtCMT3)* was expressed in *E. salsugineum* under the native *A. thaliana* promoter. In total, we assessed plants derived from six transformation events. Two of these lineages, referred to as AtCMT3-L1 and AtCMT3-L2, were propagated by single seed descent for six generations following transformation and serve as the main focus of this study (*Figure 1A*; see *Figure 1—figure supplement 1* and *Supplementary file 1* for a complete description of all plant lines and associated data described in this study). Whole genome bisulfite sequencing was completed on individual plants for each generation (numbered T1-T6) of each line to assess the impact of *AtCMT3* expression on DNA methylation. Plants expressing *AtCMT3* showed increased levels of CHG methylation in intergenic regions, associated with preexisting methylation, as well as over some gene bodies with no prior methylation (*Figure 1B* and *Figure 1—figure supplement 2*).

Differentially methylated regions (DMRs) were identified in each sequence context (CG, CHG, and CHH), comparing each generation of AtCMT3-L1 and AtCMT3-L2, and wild type (*Supplementary file 2* and *Supplementary file 3*). Consistent with AtCMT3 activity, the majority of DMRs identified were CHG DMRs, with 25,096 identified that were characterized by a median size of 407 base pairs (bp) (*Supplementary file 2*). A smaller number of CG (3,502 DMRs, median size: 189 bp) and CHH (1,763 DMRs, median size: 243 bp) were also identified (*Supplementary file 2*). The majority of CHG DMRs (>99%, in each lineage) were hypermethylated CHG DMRs in *AtCMT3* expressing lineages relative to wild type (*Supplementary file 3*). Many of the CG and CHH DMRs (between 45–72% CG and 86–98% CHH, depending on the line) were also hypermethylated DMRs in *AtCMT3* expressing lines relative to wild type, and overlapped with CHG DMRs, suggesting they result from cross-talk between CMT3-mediated CHG methylation and other pathways (*Figure 1— figure supplement 3*, *Supplementary file 3*). There was also a significant overlap of hypermethylated CHG DMRs between all individuals, suggesting that AtCMT3 was not methylating DNA randomly (*Figure 1—figure supplement 4*).

The levels of CHG methylation varied between individuals expressing *AtCMT3* (*Figure 1B*, *Supplementary file 2* and *Supplementary file 3*). Plants were transformed using the floral dip method (*Clough and Bent, 1998*), which results in random and potentially multiple transgene

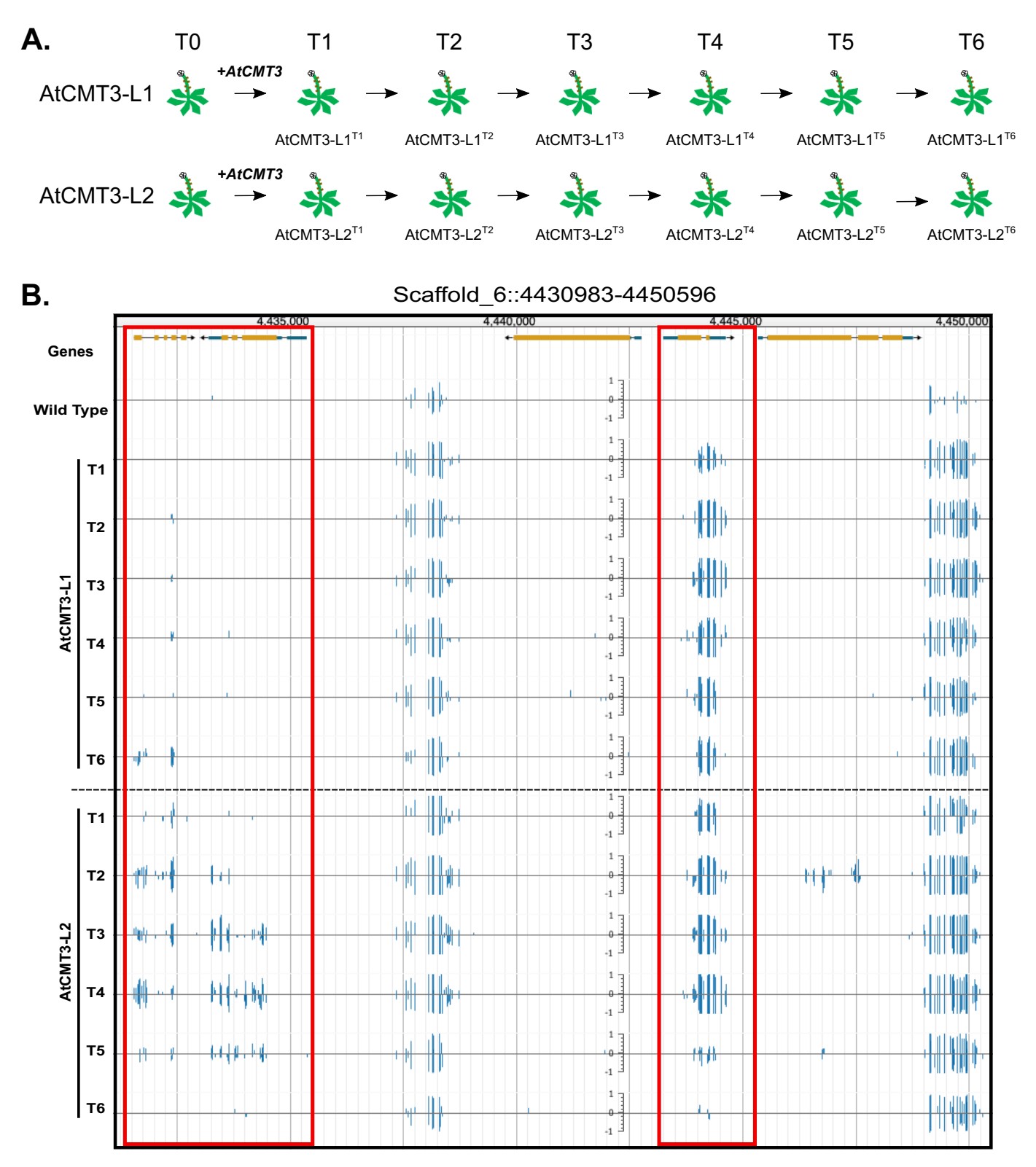

**Figure 1.** Expression of *AtCMT3* in *E.salsugineum* results in increased CHG methylation. (**A**) Schematic of the experiment. Two *E. salsugineum* lines derived from transformations with genomic *A. thaliana CMT3* were propagated by single seed decent for six generations (T1–T6). The two lines are referred to as AtCMT3-L1 and AtCMT3-L2, followed by the generation (T1–T6). For additional lines analyzed in this study see *Figure 1—figure supplement 1* and *Supplementary file 1*. (**B**) Genome browser view of CHG methylation levels derived from whole genome bisulfite sequencing. The

*Figure 1 continued on next page*

*Figure 1 continued*

image illustrates the gains in CHG methylation that occur in regions that are methylated in wild type, as well as over gene bodies with no pre-existing DNA methylation (boxed in red). Scales on tracks designate the weighted percent methylation, with 1 = 100% on the top strand and −1 = 100% on the bottom strand. Only CHG methylation is shown. For methylation in all contexts see *Figure 1—figure supplement 2*.

DOI: https://doi.org/10.7554/eLife.47891.002

The following figure supplements are available for figure 1:

**Figure supplement 1.** Diagram of all experimental *E. salsugineum* lines analyzed in this study with associated data collected.

DOI: https://doi.org/10.7554/eLife.47891.003

**Figure supplement 2.** CHG methylation increases in *AtCMT3*-expressing *E. salsugineum* lines.

DOI: https://doi.org/10.7554/eLife.47891.004

**Figure supplement 3.** CG, CHG, and CHH DMRs co-localize.

DOI: https://doi.org/10.7554/eLife.47891.005

**Figure supplement 4.** AtCMT3 expression results in gains in CHG methylation over similar regions between lineages and across generations.

DOI: https://doi.org/10.7554/eLife.47891.006

**Figure supplement 5.** Genome wide percent CHG methylation is correlated with *AtCMT3* transgene expression levels.

DOI: https://doi.org/10.7554/eLife.47891.007

insertions that can segregate out over generations. Therefore, we predicted that *AtCMT3* transgene expression likely varied between individuals, which could explain differences in CHG methylation levels. To test this, we performed RNA-sequencing in the T3 - T5 generations of AtCMT3-L1, the T3 - T6 generations of AtCMT3-L2, as well as two, T2 generation plants from an additional, independently transformed lineage (AtCMT3-L3) (*Figure 1—figure supplement 1*). The results demonstrated a significant correlation between the levels of *AtCMT3* expression and genome-wide CHG methylation levels ($R^2$ = 0.8828, p=1.669×10$^{-4}$, *Figure 1—figure supplement 5*, see *Supplementary file 4* for FPKM values). This was especially notable for two T2 generation plants of AtCMT3-L3, which had the highest *AtCMT3* expression levels (233.7 and 372.5 FPKM for AtCMT3-L3$^{T2}$ and T2b, respectively) and the highest genome-wide percent CHG methylation (27% and 32%, for T2 and T2b, respectively). These results also revealed that expression of the *AtCMT3* transgene was progressively lost in the AtCMT3-L2 lineage following the T4 generation (*Supplementary file 4*). We also examined the relationship between genome-wide CHG methylation and *AtCMT3* expression using qRT-PCR, including plants from additional lineages, and found a weaker although significant relationship ($R^2$ = 0.5932, p=0.025, *Figure 1—figure supplement 1* and *Figure 1—figure supplement 5*). Taken together, *AtCMT3* expression is likely one factor contributing to genome-wide CHG levels in these lines.

Despite lacking *CMT3*, wild-type *E. salsugineum* does exhibit residual levels of CHG methylation, likely deposited by other DNA methylation pathways such as RdDM (*Bewick et al., 2016*) (*Figure 1B* and *Figure 2A*). As the maize ortholog of CMT3 has been shown to preferentially act on hemi-methylated CHG cytosines (*Stoddard et al., 2019*), we predicted that AtCMT3 would be preferentially targeted to regions with pre-existing CHG methylation. To test this, CHG DMRs were ranked based on wild-type CHG methylation levels. Results showed that the majority of CHG DMRs (18,812/25,096 or 75%) were characterized by the presence of low levels of CHG methylation in wild type that increased upon the expression of *AtCMT3* (*Figure 2A*). Characterizing the DMRs based on overlap with genomic features revealed that 19,576 out of 25,096 (78%) total CHG DMRs overlapped with repetitive elements or intergenic regions, which are expected targets of all DNA methylation pathways (*Figure 2A*, *Supplementary file 2*). Summarizing CHG methylation over all repetitive elements confirmed that plants expressing *AtCMT3* exhibited increased CHG methylation over these regions from ~15% methylation in wild type to ~30–70% methylation in AtCMT3 lines (*Figure 2B*).

## Expression of AtCMT3 in *E. salsugineum* results in increased CHG methylation in a subset of gene bodies

In addition to methylated regions, a subset of regions (6,284 out of 25,096 (25%) CHG DMRs) that gained CHG methylation in *AtCMT3*-expressing lines had no pre-existing CHG methylation, and the majority (4,725/6,284 or 75%) of these regions overlapped annotated genes (*Figure 1B*, *Figure 2A*, and *Figure 1—figure supplement 2*, *Supplementary file 2*). Examination of percent CHG methylation over all annotated genes revealed increases in CHG methylation over gene bodies, ranging

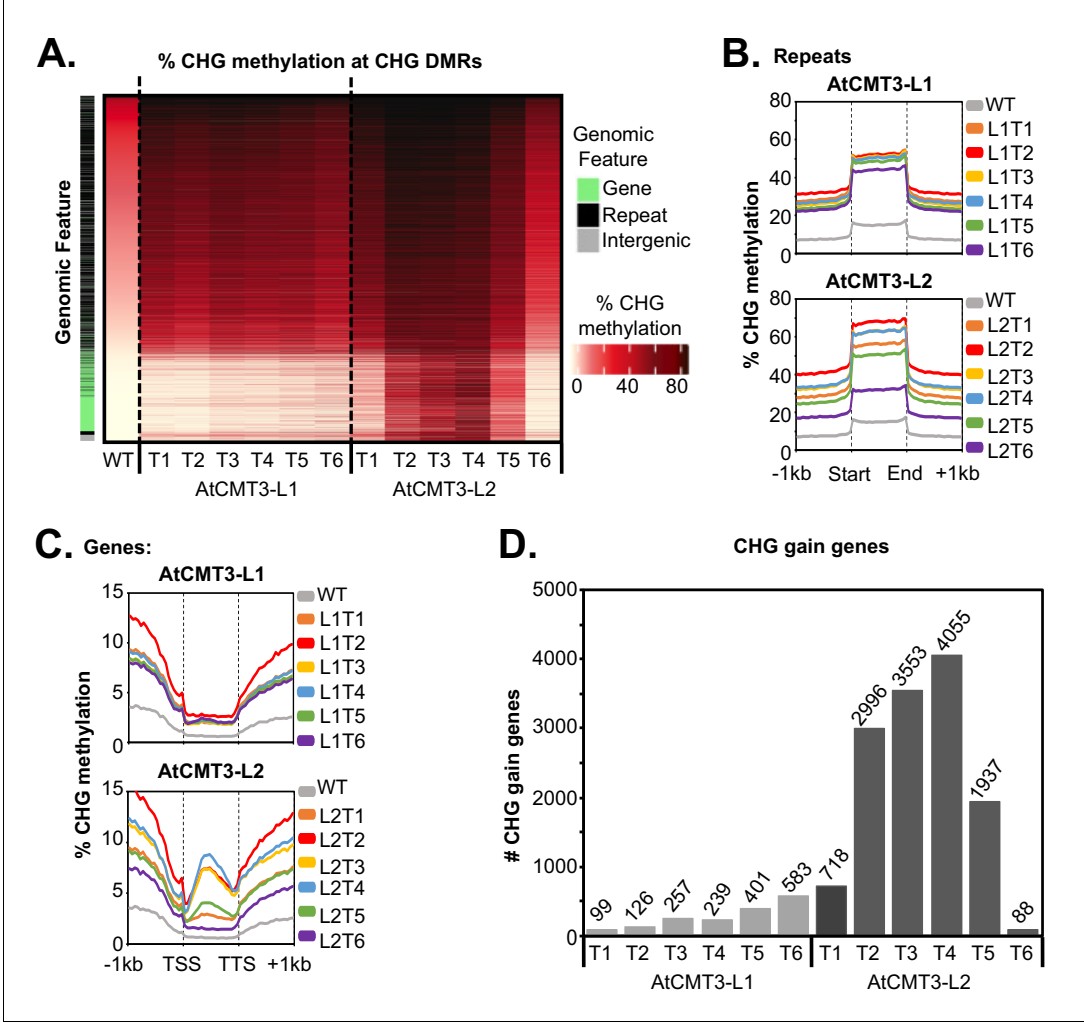

**Figure 2.** *AtCMT3* expression results in CHG methylation over repeats and a subset of gene bodies. (**A**) Heatmap of % CHG methylation over hyper-CHG differentially methylated regions (DMRs) defined by comparing all AtCMT3-L1 and L2 generations (T1–T6) to wild type. DMRs are ranked by % CHG methylation levels in wild type plants showing that the majority of gains in CHG methylation occurred over regions with pre-existing CHG methylation classified as repeats or intergenic regions. A subset of regions with no pre-existing CHG methylation, mainly classified as genes, also showed gains in CHG methylation, especially in AtCMT3-L2 individuals. See also *Supplementary file 2* and *Supplementary file 3*. (**B**) Metaplot summarizing % CHG methylation over repetitive sequences for each Line. (**C**) Metaplot summarizing % CHG methylation over gene bodies for each Line. (**D**) The number of genes gaining a minimum of 5% CHG methylation (CHG-gain genes) in each lineage. See *Supplementary file 5* for lists of CHG-gain genes.
DOI: https://doi.org/10.7554/eLife.47891.008

The following figure supplements are available for figure 2:

**Figure supplement 1.** *AtCMT3* expression results in gains in CHG methylation over similar genes between lineages and across generations.
DOI: https://doi.org/10.7554/eLife.47891.009

**Figure supplement 2.** The number of genes gaining a minimum of 5% CHG methylation (CHG-gain genes) is correlated with *AtCMT3* transgene expression levels.
DOI: https://doi.org/10.7554/eLife.47891.010

---

from ~2–10% CHG methylation on average, compared to no CHG methylation in wild-type plants (*Figure 2C*). The patterns of CHG methylation in several of the generations of the AtCMT3-L2 lineage, which exhibited higher levels of *AtCMT3* transgene expression (*Supplementary file 4*), were reminiscent of that seen for gbM (*Bewick and Schmitz, 2017*), in that the percent methylation levels

were highest towards the center of the gene bodies, decreasing towards the transcription start sites (TSS) and transcription termination sites (TTS) (*Figure 2C*).

To further characterize the genes that gained CHG methylation, genes that had less than 1% total DNA methylation in any context in wild type and that gained a minimum of 5% CHG methylation were identified in each line (CHG-gain genes) (*Figure 2D*, *Supplementary file 5*). The number of CHG-gain genes varied between individuals and lineages. In AtCMT3-L1, the number of CHG-gain genes generally increased over generational time from 99 genes in T1 to 583 genes in T6 (*Figure 2D*, *Supplementary file 5*). AtCMT3-L2 individuals were characterized by higher numbers of CHG-gain genes relative to AtCMT3-L1, with the T1 generation showing gains in 718 genes, 2,996 genes in T2, 3,553 genes in T3, and 4,055 genes in T4 (*Figure 2D*, *Supplementary file 5*). Following the T4 generation, the T5 and T6 generations of AtCMT3-L2 showed a decline in the number of CHG-gain genes, with 1,937 in T5 and only 88 genes in T6 (*Figure 2D*, *Supplementary file 5*). Despite the variation in the number of CHG-gain genes, the overlap of CHG-gain genes between all individuals was significantly higher than expected by chance (*Figure 2—figure supplement 1*). As CHG-gain genes had no prior CHG methylation, these results demonstrate a de novo methyltransferase activity of CMT3 in vivo.

We hypothesized that the variation in the number of CHG-gain genes between individuals could be related to variation in the expression levels of the *AtCMT3* transgene, similar to genome-wide CHG methylation levels. Indeed, based on RNA-seq assessment of *AtCMT3* expression, the levels of *AtCMT3* expression and number CHG-gain genes were correlated ($R^2$ = 0.7951, p=0.001) (*Figure 2—figure supplement 2*, *Supplementary file 4*). Again, two T2 generation individuals of the AtCMT3-L3 lineage, which had the highest *AtCMT3* expression (*Supplementary file 4*), also had the highest number of CHG-gain genes (5,566 and 6,346 CHG-gain genes for AtCMT3-L3$^{T2}$ and T2b, respectively) (*Supplementary file 5*). This result was also confirmed utilizing qRT-PCR (R2 = 0.9503, p=3.91×10$^{-5}$) (*Figure 2—figure supplement 2*). Therefore, the decline in the number of CHG-gain genes in the AtCMT3-L2$^{T5}$ and T6 generations is correlated with the progressive loss of *AtCMT3* transgene expression.

## CHG methylation in gene bodies is not associated with stable H3K9 methylation

CMT3 can directly bind to H3K9me2 (*Du et al., 2012*). To determine if increases of CHG methylation detected in plants expressing *AtCMT3* were correlated with H3K9me2, we conducted H3K9me2 chromatin immunoprecipitation and sequencing (ChIP-seq) in wild-type *E. salsugineum*. Consistent with CMT3 binding of H3K9me2, regions enriched for H3K9me2 in wild type showed increased CHG methylation in all lines expressing *AtCMT3* (*Figure 3A–B* and *Figure 3—figure supplement 1*).

We next sought to determine how gains in CHG methylation affected the distribution of H3K9me2 by conducting H3K9me2 ChIP-seq in the T3 and T5 generation plants for AtCMT3-L1 and AtCMT3-L2. We found plants expressing *AtCMT3* also showed enrichment for H3K9me2 across H3K9me2 ChIP peaks identified in wild type, but there were no further increases of H3K9me2 in heterochromatin, despite the increase in CHG methylation (*Figure 3A,C–F*, and *Figure 3—figure supplement 1*). In contrast to repeat regions, gains in CHG methylation over gene bodies in *AtCMT3*-expressing lines were not associated with pre-existing H3K9me2 in wild-type plants (*Figure 3A,C–F*, and *Figure 3—figure supplement 1*). Also unexpected, the establishment of CHG methylation following *AtCMT3* expression did not result in detectable H3K9me2 across CHG-gain genes (*Figure 3A,C–F*, and *Figure 3—figure supplement 1*).

CMT3 can also directly bind to H3K9me1, which distinguishes it from CMT2 (*Du et al., 2012*; *Stroud et al., 2014*). Therefore, we also conducted H3K9me1 ChIP-seq in transgenic and non-transgenic lines. Similar to findings in *A. thaliana* (*Jackson et al., 2004*), H3K9me1 was enriched over regions characterized by H3K9me2 (*Figure 3—figure supplement 2A–B*). H3K9me1 was preferentially enriched in heterochromatin relative to CHG-gain genes before or after introduction of the transgene and the H3K9me1 signal over CHG-gain genes was indistinguishable from genes that remained unmethylated (*Figure 3—figure supplement 2B–C*). Overall, we conclude that gains of CHG methylation at genic loci resulting from expression of *AtCMT3* are not associated with stable H3K9 methylation.

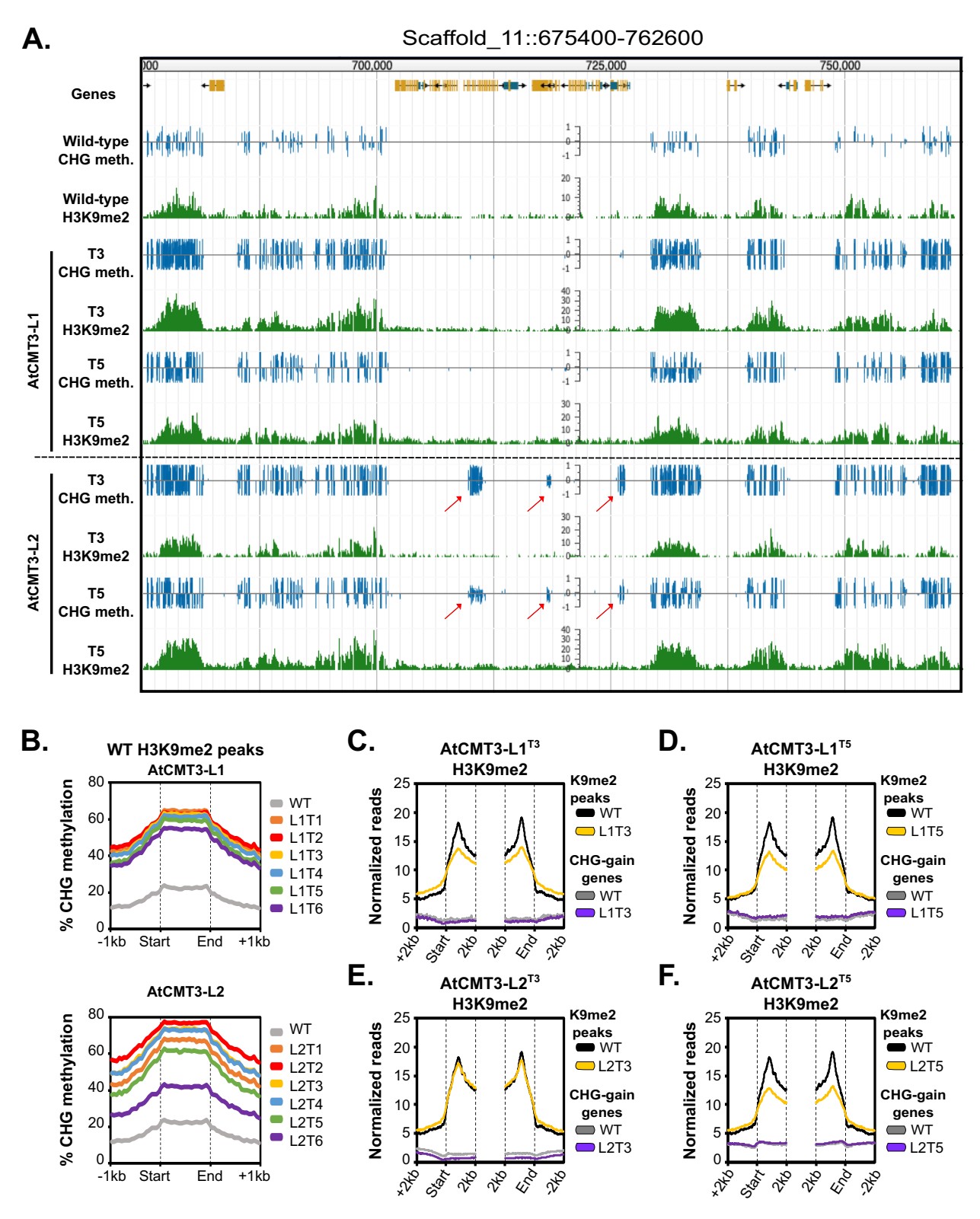

**Figure 3.** Gains in CHG methylation do not alter H3K9me2 levels or distribution. (A) Genome browser view of CHG methylation levels and H3K9me2 ChIP sequencing levels in the T3 and T5 generations of the AtCMT3-L1 and L2 lineages. Arrows indicate gains of CHG methylation over gene bodies in the AtCMT3-L2 generations that do not show H3K9me2 enrichment. Scales on methylation tracks designate the weighted percent methylation, with 1 = 100% on the top strand and −1 = 100% on the bottom strand. Scales on the H3K9me2 tracks indicate the number of mapped reads and are not

*Figure 3 continued on next page*

*Figure 3 continued*

adjusted for library size (See C-F for comparison of normalized reads). DNA methylation is only shown in the CHG context. For DNA methylation in all contexts see *Figure 3—figure supplement 1*. (B) Metaplot of % CHG methylation over H3K9me2 ChIP peaks identified in wild type plants. (C–F) Metaplot of H3K9me2 ChIP-sequencing enrichment over H3K9me2 ChIP peaks defined in wild type plants and over CHG-gain genes in AtCMT3-L1[T3] (C), AtCMT3-L1[T5] (D), AtCMT3-L2[T3] (E), and AtCMT3-L2[T5] (F). Reads were normalized to library size. See *Supplementary file 5* for lists of CHG-gain genes in each lineage.

DOI: https://doi.org/10.7554/eLife.47891.011

The following figure supplements are available for figure 3:

**Figure supplement 1.** Gains in CHG methylation do not alter H3K9me2 levels or distribution.
DOI: https://doi.org/10.7554/eLife.47891.012
**Figure supplement 2.** Gains in CHG methylation do not alter H3K9me1 levels or distribution.
DOI: https://doi.org/10.7554/eLife.47891.013

## CHG methylation in gene bodies is not associated with transcriptional silencing

CMT3-mediated CHG methylation in heterochromatin is associated with transcriptional silencing (*Bartee et al., 2001*; *Lindroth et al., 2001*; *Stroud et al., 2014*). To determine if gains in CHG methylation over gene bodies resulted in transcriptional changes, we compared expression of CHG-gain genes between wild-type and transgenic plants. In each line there was only a small proportion of CHG-gain genes (less than or equal to 10%) that showed a greater than two $\log_2$ fold change and a similar number were both down and up-regulated (*Figure 4A* and *Figure 4—figure supplement 1*, *Supplementary file 5*). There was also no relationship between the levels of CHG methylation gain and expression changes (*Figure 4A* and *Figure 4—figure supplement 1*). Additionally, we defined up- and down-regulated genes genome wide in *AtCMT3*-expressing lines (defined as greater than a two $\log_2$ fold change relative to wild type), and CHG gain genes were not significantly enriched in either up- or down-regulated genes (p>0.05, Fishers exact test) (*Supplementary file 6* and *Supplementary file 7*). We also considered that the changes in gene expression may be indirect effects of *AtCMT3* expression and assessed up- and down-regulated genes for significant enrichments in Gene Ontology (GO) biologic processes. We found that many of the enriched terms for up- and down-regulated genes were involved in abiotic stress responses (*Supplementary file 8*). However, there were no consistently identified GO-term enrichments across lineages suggesting that *AtCMT3* expression was unlikely the direct or indirect cause of these changes (*Supplementary file 8*). Lack of transcriptional changes directly related to CHG methylation is consistent with the lack of stable H3K9me2 at these loci (*Figure 3C–F*). We conclude that genic CHG methylation in *AtCMT3*-expressing lines is uncoupled from heterochromatin formation and transcriptional silencing, similar to gbM.

## Genes that gain CHG methylation in AtCMT3-expressing lines are orthologs of *A. thaliana* gbM genes and possess similar characteristics

GbM is present on conserved orthologous genes across diverse plant species (*Bewick et al., 2017*; *Niederhuth et al., 2016*; *Seymour et al., 2014*; *Takuno and Gaut, 2013*; *Takuno et al., 2016*). To determine if genes that gain CHG methylation in *AtCMT3*-expressing lineages are genes that would be predicted to have gbM based on orthology, orthologs of CHG-gain genes in *E. salsugineum* were identified in *A. thaliana*. Among the generations of AtCMT3-L1 and L2, there was a total of 4,769 CHG-gain genes identified in at least one individual (*Supplementary file 5*). Among these genes, 4,104 have an orthologous gene encoded in the *A. thaliana* genome, and, out of those, 1,526 were classified as gbM in the *A. thaliana* Col-0 accession, which is significantly more than expected by chance (p=$2.83 \times 10^{-222}$, hypergeometric test) (*Supplementary file 9*).

Relative to unmethylated genes, gbM genes are generally characterized as being constitutively expressed at moderate levels, they tend to be longer and have more exons, and have a higher frequency of CMT3-preferred CHG sites (*Bewick et al., 2016*; *Bewick et al., 2017*; *Lister et al., 2008*; *Niederhuth et al., 2016*; *Takuno and Gaut, 2012*; *Takuno and Gaut, 2013*; *Takuno et al., 2016*; *Zhang et al., 2006*). To determine whether the CHG-gain genes had similar characteristics, we compared the 4,769 CHG-gain genes identified to the remaining unmethylated *E. salsugineum* genes

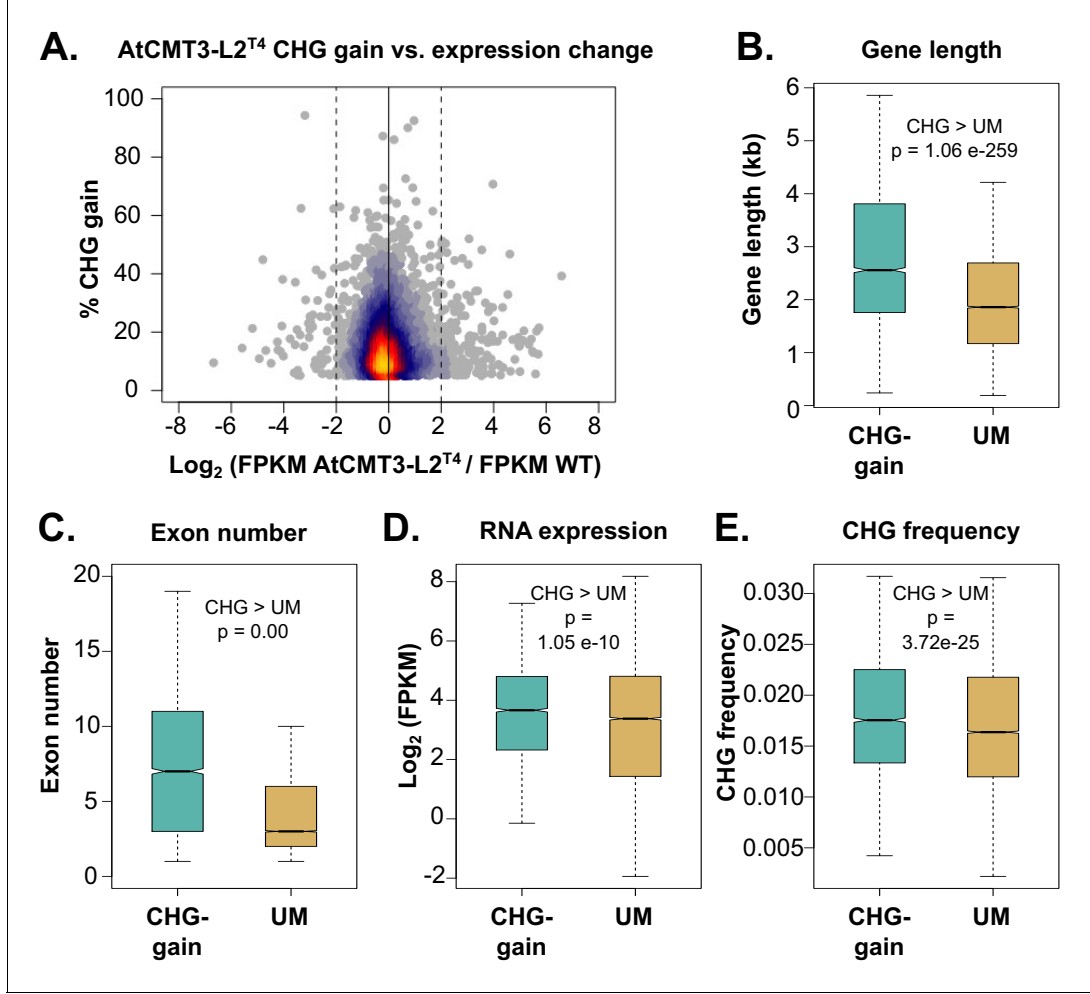

**Figure 4.** Genes that gain CHG methylation have *A. thaliana* gbM gene characteristics. (**A**) Relationship between levels of CHG methylation gain and log$_2$ fold change (FC) in expression for CHG-gain genes in AtCMT3-L2$^{T4}$ relative to wild type. Genes with zero FPKM values were removed from the analysis. See *Figure 4—figure supplement 1* for additional individuals analyzed. (**B–E**) Comparison of the (**B**) distribution of gene lengths, (**C**) number of exons, (**D**) expression levels, and (**E**) frequency of CHG sites between CHG-gain genes and unmethylated genes (UM). P-values were calculated using a Wilcoxon rank-sum test. Boxes indicate the first and third quartiles, with the center line indicating the median and notches the 95% confidence interval of the median. Whiskers show 1.0 times the interquartile range and outliers beyond this range were excluded for visualization purposes, but included in all calculations.

DOI: https://doi.org/10.7554/eLife.47891.014

The following figure supplement is available for figure 4:

**Figure supplement 1.** Changes in expression are not related to CHG methylation levels.

DOI: https://doi.org/10.7554/eLife.47891.015

(UM genes). On average, CHG-gain genes were longer, contained more exons, exhibited a more moderate, but on average higher, range of expression, and had a higher frequency of CHG cytosines relative to gene length compared to unmethylated genes (*Figure 4B–E*). Therefore, AtCMT3 in *E. salsugineum* methylates genes that are orthologs and/or characterized by similar properties of gbM loci in *A. thaliana*.

## Gains in CHG methylation over gene bodies are associated with gains in non-CHG methylation

GbM is defined as strictly CG context methylation (*Bewick and Schmitz, 2017*), which contrasts with the predominantly CHG methylation observed over gene bodies in the AtCMT3-Lines (*Figure 1B*,

*Figure 3A*, *Figure 1—figure supplement 2*, and *Figure 3—figure supplement 1*). To determine if *AtCMT3* expression resulted in non-CHG methylation over gene bodies as well, we focused on the AtCMT3-L2 lineage, which showed higher *AtCMT3* expression levels and CHG-gain genes relative to AtCMT3-L1. We analyzed all CHG DMRs that overlapped CHG-gain genes identified in the T4 generation (4,312 CHG DMRs overlapping 4,055 CHG-gain genes) (*Supplementary file 2* and *Supplementary file 5*), and determined the levels of CG, CHG, and CHH methylation over these regions (*Figure 5A*). Re-focusing the analysis to the level of DMRs over genes, rather than whole genes, revealed some residual methylation present on genes below our original cutoff of 1% gene-wide methylation. This residual methylation was almost exclusively in the CG context and is consistent with CMT3 preferentially localizing to methylated DNA (*Figure 5A*). Importantly, 2,094 of the

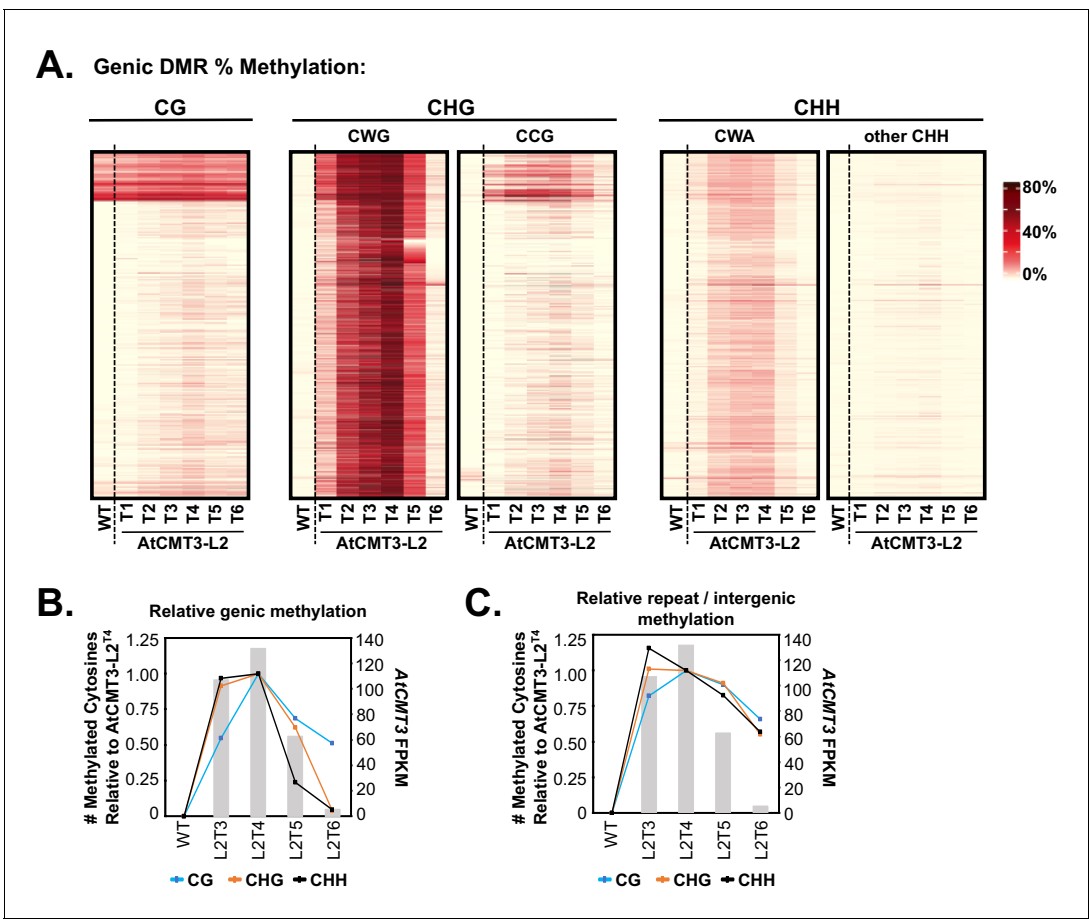

**Figure 5.** CHG methylation over gene bodies is associated with gains in non-CHG methylation. (A) Heatmap of % methylation levels across CHG DMRs overlapping the CHG-gain genes in AtCMT3-L2$^{T4}$ divided into selected trinucleotide contexts. See *Figure 5—figure supplement 1A* for further parsing of the data into all 16 possible trinucleotide contexts. (B) Assessment of the relationship of genic cytosine methylation to *AtCMT3* expression across AtCMT3-L2 generations. Line plots show the number of methylated cytosines in each context relative to AtCMT3-L2$^{T4}$ across CHG DMRs overlapping CHG-gain genes identified in AtCMT3-L2$^{T4}$. Bar plots show the expression of the *AtCMT3* transgene. For further parsing of the CHG and CHH contexts into CWG vs. CCG and CWA vs. other CHH, see *Figure 5—figure supplement 1B–C*. (C) Assessment of the relationship of repeat and intergenic cytosine methylation to *AtCMT3* expression across AtCMT3-L2 generations. Line plots show the number of methylated cytosines relative to AtCMT3-L2$^{T4}$ across hyper CHG DMRs identified in AtCMT3-L2$^{T4}$ that overlap repeats or intergenic regions. Bar plots show the expression of the *AtCMT3* transgene.

DOI: https://doi.org/10.7554/eLife.47891.016

The following figure supplements are available for figure 5:

**Figure supplement 1.** CHG methylation over gene bodies is associated with gains in non-CHG methylation.
DOI: https://doi.org/10.7554/eLife.47891.017

**Figure supplement 2.** AtCMT3-induced genic CG methylation is maintained at higher levels than background following loss of *AtCMT3* expression.
DOI: https://doi.org/10.7554/eLife.47891.018

4,312 regions had no detectable CHG methylation in wild type and 597 of the 4,312 had no methylation in any sequence context (see also *Figure 5A*, *Figure 1—figure supplement 2*, and *Figure 3—figure supplement 1*).

As observed previously, the major gains in genic methylation in AtCMT3-Lines occurred in the CHG contexts (*Figure 5A*). Further dividing CHG methylation into CWG and CCG contexts revealed that the CHG methylation was highly enriched for CWG methylation, consistent with the preferred substrates of CMT3 (*Gouil and Baulcombe, 2016*; *Stoddard et al., 2019*) (*Figure 5A*). Widespread, but lesser gains were also identified for CHH cytosines (*Figure 5A*). The CHH methylation was almost exclusively in the CWA context, which is indicative of CMT2 activity and not RdDM (*Gouil and Baulcombe, 2016*), suggesting that CMT3 activity likely leads to the recruitment of CMT2. Gains in genic CG methylation were comparatively lower than the other sequence contexts, but did occur appreciably over most loci, including regions with no prior CG methylation (*Figure 5A*). Further parsing of the data into all 16 trinucleotide contexts did not reveal any additional trends in the patterns of methylation (*Figure 5—figure supplement 1A*).

## Ectopic genic CG methylation is preferentially maintained following loss of AtCMT3 expression

Establishment of methylation in non-CHG sequence contexts over CHG-gain genes is consistent with CMT3 activity recruiting additional methyltransferase pathways. The natural transgene silencing that occurred in the AtCMT3-L2 lineage following the T4 generation provided an opportunity to test this hypothesis and determine whether non-CHG methylation remained following loss of *AtCMT3* expression. To do so, we analyzed AtCMT3-L2 individuals for which we had both RNA-seq and whole genome bisulfite sequencing data (wild type, AtCMT3-L2 T3-T6) relative to the T4 generation, which showed the highest number of CHG-gain genes. The number of newly methylated cytosines in each sequence context in the CHG DMRs overlapping the 4,055 CHG-gain genes identified in AtCMT3-L2$^{T4}$ were identified for each individual (same regions as *Figure 5A*; cytosines analyzed were corrected for coverage in all lines). Then the ratio of methylated cytosines was calculated relative to the T4 generation.

Results showed that the proportional CHG methylation levels on genes mirrored *AtCMT3* expression levels, with a slight increase in proportional methylation from 92% in AtCMT3-L2$^{T3}$ to 100% (by definition) in AtCMT3-L2$^{T4}$ (*Figure 5B*). Following the T4 generation, and consistent with progressive silencing of the *AtCMT3* transgene, the number of CHG cytosines methylated showed a marked decrease, with only 62% of T4 levels remaining methylated in the T5 generation and 4% remaining methylated in the T6 generation where *AtCMT3* expression was greatly reduced (*Figure 5B*). CHH context cytosines showed a similar trend with 97% cytosines methylated in the T3 relative to the T4 generation, followed by a steep decline, with only 24% and 5% methylated remaining in the T5 and T6 generations, respectively, relative to the T4 (*Figure 5B*). Further dividing CHG and CHH cytosines into CWG vs. CCG or CWA vs. other CHH contexts did not reveal any differences in these trends (*Figure 5—figure supplement 1B–C*).

CG context cytosines showed a more substantial increase from the T3 to T4 generation, with CG cytosines in the T3 generation at 54% of the levels measured in T4 (*Figure 5B*). Also contrasting with CHG and CHH methylation, following the T4 generation, the proportion of methylated CG sites remaining following the progressive *AtCMT3* silencing was relatively high and much more stable, with 68% in T5% and 51% in T6 of the CG cytosines remaining methylated, relative to T4 (*Figure 5B*).

The results of the analysis of genic methylation for AtCMT3-L2 contrast with the identical analysis conducted on AtCMT3-L1, which did not show evidence of silencing of the *AtCMT3* transgene. In this case, the relative methylation levels were maintained in all sequence contexts across generations (*Figure 5—figure supplement 1D*). Furthermore, to verify that the remaining genic CG methylation in AtCMT3-L2$^{T6}$ did not result from residual AtCMT3 activity, we also completed an alternative approach where we crossed the AtCMT3-L1$^{T5}$ transgene expressing line to wild type (non-transgenic) to segregate out the transgene. We analyzed two F2 progeny from this cross in which the transgene was segregated out and one F2 progeny that still contained the transgene (*Figure 1—figure supplement 1*). In the F2 progeny that did not encode the transgene, the ectopic genic CHG and CHH methylation that was present in the in the transgenic parent was lost but the genic CG methylation was maintained, similar to the result from AtCMT3-L2 when the transgene was silenced

(*Figure 5—figure supplement 1E*). In contrast, the F2 progeny that still encoded the transgene maintained similar levels of genic methylation in all contexts as the transgenic parent (*Figure 5—figure supplement 1E*).

Importantly, the levels of newly methylated CG cytosines on CHG-gain genes in both the lines that lost *AtCMT3* through silencing or crossing out were higher than background due to bisulfite non-conversion, estimated by comparing them to randomly sampled, un-methylated genes that did not gain CHG methylation (*Figure 5—figure supplement 2A–B*). The genic CG methylation is also unlikely to have occurred independently of AtCMT3, as the gains in genic CG methylation at these loci are also higher that those found by comparing an additional, non-transgenic *E. salsugineum* accession (Yukon) (*Figure 5—figure supplement 2A–B*). Thus, genic CG methylation showed a lesser dependency on *AtCMT3* expression and was preferentially maintained following loss of transgene expression, either through natural silencing or crossing out the transgene, consistent with maintenance of CG methylation by other methyltransferases recruited by initial AtCMT3 activity.

## AtCMT3 preferentially methylates heterochromatin relative to genes

The majority CHG DMRs resulting from *AtCMT3* expression were annotated as repeats or intergenic regions with prior methylation in wild type (~75% compared to ~25% genic loci with no prior methylation) (*Figure 2A*). Furthermore, the levels of CHG methylation in *AtCMT3* expressing lines were much higher on repeats relative to genes (30–70% CHG methylation on repeats compared to 2–10% on genes) (*Figure 2B–C*). Therefore, we next sought to determine whether this difference was a result of AtCMT3 preferentially methylating heterochromatic loci relative to genic loci. We reasoned that favorable activity of AtCMT3 on heterochromatin could be revealed in lines where *AtCMT3* transgene expression was reduced, as they would be expected to show a greater proportional loss of CHG methylation over genic regions relative to heterochromatic loci when compared to lines with high *AtCMT3* expression.

To test this, we performed the same analysis reported in *Figure 5B*, except instead of genic loci, we focused on regions defined as hyper-CHG DMRs in AtCMT3-L2$^{T4}$ that did not overlap genes (i.e. repeats and intergenic loci). Results showed that the relative number of methylated cytosines was much more robustly maintained in the AtCMT3-L2 T5 and T6 generations at these loci relative to genic loci following loss of *AtCMT3* transgene expression (compare *Figure 5B and C*). This was especially evident in the T6 generation that showed a $-4.6 \log_2$ fold change in *AtCMT3* expression relative to AtCMT3-L2$^{T4}$ (*Figure 5C*). Despite substantial loss of transgene expression, 55% and 57% of CHG and CHH cytosines, respectively, remained methylated in the T6 generation relative to T4 across repeats and intergenic loci (*Figure 5C*). In contrast, only 4% of CHG and 5% of CHH cytosines remaining methylated across genes (*Figure 5B*). Methylated CG cytosines were also slightly more robust to loss of *AtCMT3* transgene expression, dropping to 66% of T4 levels over repeats compared to 51% over genes in the T6 generation (*Figure 5B–C*).

An alternative explanation for these results is that methylation induced by *AtCMT3* expression is more readily maintained by other methylation pathways in heterochromatin relative to genes, as other methyltransferases preferentially target these regions. To consider this possibility, we also analyzed the relative maintenance of DNA methylation in heterochromatin in lines in which the transgene was removed via crossing to wild type. In the F2 progeny of the AtCMT3-L1T5 X wild type cross that no longer encoded the transgene, both CG and CHH methylation were maintained at relatively similar levels as the transgenic parent, in contrast to CHG methylation that was only maintained at ~25% of the levels of the transgenic parent (*Figure 5—figure supplement 1F*). This is consistent with the possibility that AtCMT3-induced CG and CHH methylation, and to a lesser extent CHG methylation, can be perpetuated following loss of *AtCMT3* in heterochromatin in preference to genes by other methylation pathways. However, the relative levels of CHG methylation in particular are lower than those detected over heterochromatin for lines in which the transgene was silenced, consistent with residual AtCMT3 preferentially targeting heterochromatin in these lines (*Figure 5C*). We conclude that AtCMT3 preferentially targets heterochromatin and does not readily methylate genic loci until expressed at high levels.

## Discussion

We have provided experimental evidence that CMT3 can initiate epimutations in the form of gene body CG methylation, which are maintained over generational time, even after loss of *AtCMT3* expression. This finding has provided new insights into CMT3 localization and function by showing CMT3 is associated with de novo DNA methylation activity in vivo at genic loci lacking prior DNA methylation. The results also revealed a mechanism for the establishment of gbM that is consistent with the hypothesis that gbM is a passive effect of self-reinforcing positive feedback loops inherent to the heterochromatin machinery.

The natural loss of *CMT3* in *E. salsugineum* and other species associated with the loss of gbM (*Bewick et al., 2016*; *Niederhuth et al., 2016*), combined with prior work demonstrating that CMT3 targeting and activity requires binding to H3K9me (*Du et al., 2012*; *Stoddard et al., 2019*), suggests that both CMT3 and histone methyltransferases are important in the establishment of gbM. Furthermore, a plausible means for CMT3 activity to recruit additional methyltransferase pathways that deposit methylation in additional sequence contexts, such as CMT2, is indirectly through the promotion of H3K9me2 (*Du et al., 2015*). We therefore propose that both enzymes work in concert to provide de novo methylation of transcribed loci to initially establish gbM through the model described in *Figure 6*, which expands on prior models (*Inagaki and Kakutani, 2012*).

In this model, nucleosomes possessing H3K9me are on rare occasions incorporated into transcribed genes. Initially, genic histone methylation could be restricted to H3K9me1, which is bound by CMT3 and not CMT2 (*Stroud et al., 2014*), and could potentially explain the phylogenetic correlation between encoding *CMT3* and the presence of gbM across plant species (*Bewick et al., 2016*; *Niederhuth et al., 2016*). Most H3K9me associated nucleosomes are efficiently removed via the histone de-methylase, INCREASED BONSAI METHYLATION 1 (IBM1), in a transcription coupled mechanism. In *A. thaliana*, gbM loci are devoid of H3K9me2 due to the activity of IBM1, which prevents the establishment of H3K9me2 at gbM loci through a mechanism that requires active transcription (*Inagaki et al., 2010*; *Saze et al., 2008*). It is notable that encoded in the *E. salsugineum* genome are several expressed orthologs of the *A. thaliana IBM1* (*Supplementary file 10*), which could destabilize H3K9me and plausibly explain the lack of detection of H3K9me over genes that gain CHG methylation (*Figure 3*, *Figure 3—figure supplement 1* and *Figure 3—figure supplement 2*). However, CMT3 binding to H3K9me may transiently stabilize H3K9me2 through the establishment of de novo CHG methylation, which activates the feedback loop between DNA and histone methyltransferases. This contrasts with CMT3 activity at heterochromatin, where CMT3 preferentially methylates hemi-methylated CWG sites and is re-enforced by additional methylation pathways. As de novo methylation is a less favored activity of CMT3, this process is predicted to occur rarely, but can be promoted with high levels of CMT3 expression. Transient stabilization of H3K9me2 recruits additional methyltransferases, including CMT2, which establish DNA methylation in additional sequence contexts. Finally, removal of H3K9me2 by IBM1 disrupts the feedback loop between H3K9me2 and CMTs resulting in the loss of non-CG methylation following DNA replication. CG methylation is maintained, however, due to the preferential recruitment of CG maintenance methyltransferases to hemi-methylated sites following DNA replication (*Figure 6*).

From a mechanistic standpoint, it is most parsimonious to conclude that gbM is a passive byproduct inherent to the function of CMT3 in the maintenance of heterochromatin. The presence of pathways that work to uncouple gbM from transcriptional silencing, such as the IBM1 pathway, further support this line of reasoning, as they may have evolved to counteract negative consequences of 'spillage' of the heterochromatin machinery into genic space. Why, then, are some genes consistently characterized by body methylation across species and others not? It is telling that the genes that gain methylation in *E. salsugineum* are homologs of gbM genes in *A. thaliana* and/or retain similar characteristics including gene length, expression profile, and relative frequency of CHG sites. Rather than an exact determinant of gbM status, it is likely that gene length and constitutive expression contribute to the exposure of a locus to incorporation of H3K9me1/2 nucleosomes, which, combined with the frequency of CMT3-preferred CWG sites and CMT3 levels, make a gene susceptible to methylation by CMT3 in a probabilistic manner. Under this model, gbM can be thought of as an epigenetic scar resulting from transient localization of the heterochromatin machinery that is likely to be present on a given gene as a function of these factors. This model does not exclude functional consequences of gbM, but it also does not require them.

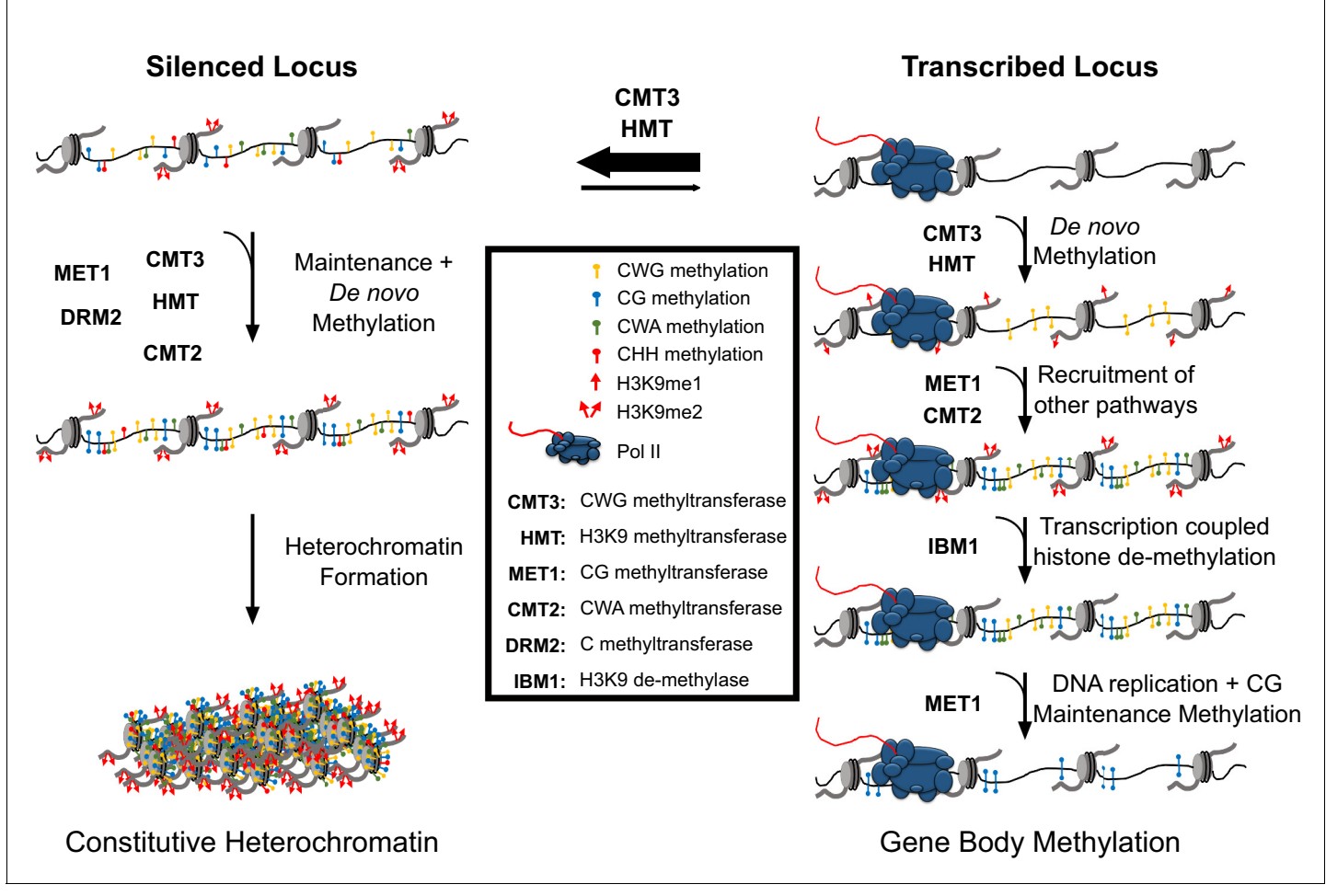

**Figure 6.** Hypothetical model for CMT3 establishment of gbM. The activity of CMT3 and histone methyltransferases (HMTs) maintains CWG methylation and H3K9me2, respectively, and is most readily detected at silenced loci. At silenced loci, methylation by CMT3 and HMTs is reinforced by other methyltransferase pathways, including MET1 (CG methylation), CMT2 (CWA methylation), and DRM2 (methylation in all contexts), which maintain constitutive heterochromatin. In contrast to silenced loci, CMT3 and HMTs can only transiently establish de novo CWG methylation and H3K9me at transcribed loci characterized by gbM. This process may be initiated by incorporation of H3K9me1 nucleosomes, which are bound exclusively by CMT3 and not CMT2 and normally removed by the H3K9 de-methylase IBM1 in a transcription coupled mechanism. However, on rare occasions CMT3 may bind H3K9me1 located in genes and de novo methylate CWG cytosines. De novo methylation of CMT3 is not a favored activity of the enzyme, so this happens only very rarely or when CMT3 is expressed at high levels. This temporally stabilizes H3K9me due to the self-reinforcing feedback loop between histone and DNA methyltransferases. Transient stabilization of H3K9me promotes H3K9me2 that can then recruit additional methyltransferases including MET1 and CMT2 to methylate CG and CWA cytosines, respectively. Heterochromatin formation is inhibited, however, through eventual removal of H3K9me by IBM1. Loss of H3K9me and/or loss of available CMT3 results in the failure of maintenance of DNA methylation in all contexts except CG following DNA replication. CG methylation is maintained due to the preferential targeting of the CG maintenance methyltransferase, MET1, to hemi-methylated cytosines following replication.

DOI: https://doi.org/10.7554/eLife.47891.019

# Materials and methods

## Key resources table

| Reagent type (species) or resource | Designation | Source or reference | Identifiers | Additional information |
|---|---|---|---|---|
| Strain, strain background (*Agrobacterium tumefaciens*) | C58C1 | other | | Dr. Robert Schmitz (University of Georgia) |

*Continued on next page*

*Continued*

| Reagent type (species) or resource | Designation | Source or reference | Identifiers | Additional information |
|---|---|---|---|---|
| Strain, strain background (*Eutrema salsugineum*) | Shandong | https://www.arabidopsis.org | CS22504 | |
| Strain, strain background (*E. salsugineum*) | Shandong AtCMT3 lineages | This paper | | Dr. Robert Schmitz (University of Georgia) |
| Strain, strain background (*E. salsugineum*) | Yukon | https://www.arabidopsis.org | CS22664 | |
| Antibody | anti-H3K9me2 | Cell Signaling Technology | Cat# 9753 s; RRID: AB_659848 | Polyclonal, 5 µg |
| Antibody | anti-H3K9me1 | Abcam | Cat# 8896; RRID: AB_732929 | Polyclonal, 5 µg |
| Recombinant DNA reagent | pEarleyGate 302 pAtCMT3::gAtCMT3 | PMID: 23021223 | | |
| Peptide, recombinant protein | T4 DNA Ligase | NEB | Cat# M0202 | |
| Peptide, recombinant protein | Klenow Fragment | NEB | Cat# M0210 | |
| Peptide, recombinant protein | Phusion DNA Polymerase | NEB | Cat# M0530 | |
| Peptide, recombinant protein | SuperScript III Reverse Transcriptase | Invitrogen | Cat# 18080044 | |
| Commercial assay or kit | Qiagen DNeasy Plant Mini Kit | Qiagen | Cat# 69106 | |
| Commercial assay or kit | EZ DNA-methylation Gold Kit | Zymogen | Cat# D5006 | |
| Commercial assay or kit | AMPure beads | Beckman Coulter | Cat# A63880 | |
| Commercial assay or kit | NEXTFLEX Bisulfite Sequencing Library Prep Kit | Bioo Scientific | Cat# NOVA-5119–01 | |
| Commercial assay or kit | KAPA HiFi Uracil+ | Roche | Cat #07959079001 | |
| Commercial assay or kit | Direct-Zol RNA Mini-prep plus | Zymogen | Cat# R2071 | |
| Commercial assay or kit | Illumina TruSeq mRNA Stranded Library Kit | Illumina | Cat# 20020594 | |
| Commercial assay or kit | Protein A Dynabeads | Invitrogen | Cat# 10001D | |
| Commercial assay or kit | LightCycler 480 SYBR green master mix | Roche | Cat# 04707516001 | |
| Chemical compound, drug | Silwet L-77 | Phyto Technology Laboratories | Cat#:S7777 | |
| Chemical compound, drug | Pierce Protease Inhibitors | ThermoFisher | Cat# A32963 | |
| Chemical compound, drug | NEBNext dA-Tailing Reaction Buffer | NEB | Cat# B6059 | |
| Chemical compound, drug | proteinase K | ThermoFisher | Cat# 26160 | |
| Software, algorithm | methylpy | PMID: 26030523 | | https://github.com/yupenghe/methylpy |
| Software, algorithm | cutadapt v1.9.dev1 | DOI: https://doi.org/10.14806/ej.17.1.200 | RRID:SCR_011841 | https://cutadapt.readthedocs.io/en/stable/ |

*Continued on next page*

*Continued*

| Reagent type (species) or resource | Designation | Source or reference | Identifiers | Additional information |
|---|---|---|---|---|
| Software, algorithm | bowtie 2.2.4 | PMID: 22388286 | RRID:SCR_005476 | http://bowtie-bio.sourceforge.net/bowtie2/index.shtml |
| Software, algorithm | Intervene v.0.6.1 | PMID: 28569135 | | https://intervene.read thedocs.io/en/latest/ |
| Software, algorithm | Bedtools v2.27.1 | PMID: 20110278 | RRID:SCR_006646 | https://bedtools.read thedocs.io/en/latest/ |
| Software, algorithm | HISAT2 v2.0.5 | PMID: 25751142 | RRID:SCR_015530 | https://ccb.jhu.edu/software/hisat2/index.shtml |
| Software, algorithm | StringTie v1.3.3b | PMID: 25690850 | RRID:SCR_016323 | https://ccb.jhu.edu/software/stringtie/#pub |
| Software, algorithm | HOMER 4.10 | PMID: 20513432 | RRID:SCR_010881 | http://homer.ucsd.edu/homer/ |
| Software, algorithm | Trimmomatic v0.33 | PMID: 24695404 | RRID:SCR_011848 | http://www.usadellab.org/cms/?page=trimmomatic |
| Software, algorithm | Bowtie v1.1.1 | PMID: 19261174 | RRID:SCR_005476 | http://bowtie-bio.sourceforge.net/index.shtml |
| Software, algorithm | SAMtools v1.2 and v0.1.19 | PMID: 19505943 | RRID:SCR_00210 | http://samtools.sourceforge.net |
| Software, algorithm | R v3.44 | other | RRID:SCR_001905 | https://www.r-project.org |

## Contact for reagent and resource sharing

Further information and requests for resources and reagents should be directed to and will be fulfilled by Robert J. Schmitz (schmitz@uga.edu).

## Experimental models and subject details

*Eutrema salsugineum* Shandong ecotype was grown on soil at 21°C in long day conditions (16 hr light, 8 hr dark). Plant transgenesis was conducted using the floral dip method (*Clough and Bent, 1998*). The pEarleyGate 302 vector containing genomic sequence of *Arabidopsis thaliana CMT3*, including the native promoter, published by *Du et al. (2012)* was transformed into *Agrobacterium tumefaciens* strain C58C1. Bacteria were grown for 2 days at 30°C in 200 ml cultures containing gentamicin (25 µg/mL), kanamycin (50 µg/mL), and rifampicin (50 µg/mL) and pelleted by centrifugation at 4°C. Bacterial pellets were resuspended in 5% sucrose and 0.03% Silwet L-77 (Phyto Technology Laboratories) and used to dip open *E. salsugineum* inflorescences. Transgenic plants were selected for using Finale (BASTA, Bayer).

## Whole genome bisulfite sequencing

Whole genome bisulfite sequencing libraries were generated based on methods described in *Urich et al. (2015)*. DNA was extracted from cauline leaves of individual plants flash-frozen in liquid nitrogen using the Qiagen DNeasy Kit according to the manufacturer's instructions. DNA was fragmented via sonication to a peak size of ~200 bp and further size selected using AMPure beads (Beckman Coulter) to between 150 bp and 500 bp. Fragment end repair was performed using End-It from Lucigen incubated at room temperature for 45 min, followed by purification using AMPure beads. Next, A-tailing was conducted using Klenow Fragment from NEB in NEBNext dA-tailing reaction buffer at 37°C for 30 min, followed by purification with AMPure beads and Illumina indexed adapter (NEXTFLEX Bisulfite-Seq Barcodes) ligation using T4 DNA ligase from NEB. Ligation was conducted at 16°C for 16 hr. Ligation products were purified twice using AMPure beads and bisulfite converted using the EZ DNA Methylation-Gold Kit from Zymogen. Bisulfite converted DNA was then amplified using KAPA HiFi Uracil + and universal primers with the following parameters: 95°C for 2 min, 98°C for 30 s, 8 cycles of 98°C for 15 s, 60°C for 30 s, 72°C for 4 min, and a final extension time of 72°C for 10 min. PCR products were purified using AMPure beads and sequenced with an Illumina

NextSeq500 instrument by the Georgia Genomics and Bioinformatics Core. Adapters and primers used were those provided in the NEXTFLEX Bisulfite Sequencing Library Prep Kit (Bioo Scientific).

## RNA-sequencing

Total RNA was extracted from cauline leaves of individual plants using the Direct-Zol RNA Mini-prep plus kit from Zymogen according to the manufacturer's instructions. Sequencing libraries were prepared from 1.3 µg input RNA with the Illumina TruSeq mRNA Stranded Library Kit according to the manufacturer's instructions, except all volumes were reduced to 1/3 of the recommended quantity. Sequencing was completed using an Illumina NextSeq500 instrument by the Georgia Genomics and Bioinformatics Core.

## Chromatin immunoprecipitation and sequencing (ChIP-seq)

ChIP was conducted based on the protocol described in *Schubert et al. (2006)*. Cauline leaves were harvested from individual plants and submerged in cross-linking buffer (10 mM Tris-HCl pH 8, 1 mM EDTA, 0.4 M sucrose, 100 mM PMSF, 1% formaldehyde). Tissues were vacuum infiltrated for 5 min at 85 kPa, followed by release of vacuum and 10 additional minutes at 85 kPa. Crosslinking was quenched with the addition of glycine to a concentration of 100 mM followed by 5 min under vacuum at 85 kPa. Tissues were then washed five times in water, flash frozen in liquid nitrogen, and ground to a fine powder with mortar and pestle. Powder was suspended in 10 mls extraction buffer 1 (0.4 M sucrose, 10 mM Tris-HCl pH 8, 10 mM MgCl$_2$, 5 mM BME, 0,1 mM PMSF, 1 mM EDTA, one tab/10 ml Pierce Protease Inhibitors (ThermoFisher)). The suspension was filtered through 2 layers of Miracloth to enrich nuclei and pelleted by centrifugation for 20 min at 4000 rpm at 4°C. Pellets were resuspended in 1 ml extraction buffer 2 (0.25 M sucrose, 10 mM Tris-HCl pH 8, 10 mM MgCl$_2$, 1% Triton X-100, 1 mM EDTA, 5 mM BME, 0.1 mM PMSF, one tab/10 ml protease inhibitors) and centrifuged for 10 min at 12,000 g at 4°C. Pellets were then suspended in 300 µl extraction buffer 3 (1.7 M sucrose, 10 mM Tris-HCl pH 8, 0.15% Triton X-100, 2 mM MgCl$_2$, 1 mM EDTA, 5 mM BME, 0.1 mM PMSF, one tab/10 ml protease inhibitors) and layered on top of 300 µl extraction buffer 3. Samples were then centrifuged for 1 hr at 16,000 g at 4°C, supernatant was removed, and chromatin pellets were resuspended in 100 µl nuclei lysis buffer (50 mM Tris-HCl pH 8, 10 mM EDTA, 1% SDS, 0.1 mM PMSF, one tab/10 ml protease inhibitors). Chromatin was fragmented via sonication to a fragment size of ~200 base pairs and debris were removed by centrifugation at 16,000 g for 5 min at 4°C. 10 µl of the supernatant was removed for input controls and the remaining supernatant was diluted 1:10 in ChIP dilution buffer (1.1% Triton X-100, 1.2 mM EDTA, 16.7 mM Tris-HCl pH 8, 167 mM NaCl, 0.1 M PMSF, one tab/10 ml protease inhibitors).

Protein A Dynabeads (Invitrogen) were prepared by washing 25 µl beads three times with 1 ml ChIP dilution buffer (1.1% Triton X-100, 1.2 mM EDTA, 16.7 mM Tris-HCl pH 8, 167 mM NaCl). Beads were then resuspended in 100 µl ChIP dilution buffer with 5 ug anti-H3K9me2 (Cell Signaling Technology, Cat. # 9753 s) or anti-H3K9me1 (Abcam # 8896) added. Antibodies were bound to beads at 4°C with end over end rotation for 3 hr. Beads were then washed three times with ChIP dilution buffer (1.1% Triton X-100, 1.2 mM EDTA, 16.7 mM Tris-HCl pH 8, 167 mM NaCl, 0.1 M PMSF, one tab/10 ml protease inhibitors) and resuspended in the diluted chromatin samples from above. Samples were incubated with rotation over night at 4°C.

Following incubation, beads were washed twice with 1 ml each of low salt wash buffer (150 mM NaCl, 0.1% SDS, 1% TritonX-100, 2 mM EDTA, 20 mM Tris-HCl pH 8), high salt wash buffer (500 mM NaCl, 0.1% SDS, 1% TritonX-100, 2 mM EDTA, 20 mM Tris-HCl pH 8), and LiCl wash buffer (0.25 LiCl, 1% NP40, 1% sodium deoxycholate, 1 mM EDTA, 10 mM Tris-HCl pH 8). Beads were then washed one time with 1 ml TE buffer (10 mM Tris-HCl pH 8, 1 mM EDTA) and resuspended in 250 µl elution buffer (1% SDS, 0.1 M NaHCO$_3$). Samples were eluted with incubation at 65°C for 15 min with gentle agitation. The supernatant was removed and saved and the elution was repeated with an additional 250 µl elution buffer. Supernatants were then combined and 20 µl 5 M NaCl was added. 500 µl elution buffer and 20 µl 5 M NaCl were also added to the input controls. Crosslinks were reversed over night at 65°C. Following crosslink reversal, 10 µl of 0.5 M EDTA, 20 µl 1 M Tris-HCl (pH 6.5), and 2 µl of 10 mg/ml proteinase K (ChIP grade, Thermo Fisher Scientific) was added to each sample and incubated at 45°C for 1 hr. DNA was extracted with phenol/chloroform/isoamyl alcohol (25:24:1) and resuspended in water.

ChIP sequencing libraries were prepared by conducting end repair, A-tailing, and adaptor ligation steps identical to those described for bisulfite sequencing library preparation, except that with the substitution of Illumina TruSeq adaptors and indexed primers. Libraries were amplified with Phusion DNA Polymerase (NEB) with the following parameters: 95°C for 2 min, 98°C for 30 s, 15 cycles of 98°C for 15 s, 60°C for 30 s, 72°C for 4 min, and a final extension step of 72°C for 10 min. Libraries were sequenced on an Illumina NextSeq500 instrument by the Georgia Genomics and Bioinformatics Core.

## qRT-PCR

RNA was extracted from cauline leaves of individual plants using the Direct-Zol RNA Mini-prep plus kit from Zymogen according to the manufacturer's instructions. Synthesis of cDNA was completed using SuperScript III with random hexamers (Invitrogen) according to the manufacturer's instructions. Real time qRT-PCR was conducted using LightCycler 480 SYBR green master mix in a Light Cycler 480 instrument (Roche). Primers used include: AtCMT3 qRT-PCR FP: TGGTTTGAACCTCGTCAC TAAA; AtCMT3 qRT-PCR RP: CGTTTGTCTCTGGGTGGTTAT; EsTUB4 qRT-PCR FP: CCTCCATA TCCAAGGCGGTC; EsTUB4 qRT-PCR RP: GTACTGGCCGGTGTGATCAA.

## Whole genome bisulfite sequencing mapping and analyses

WGBS data were processed using 'single-end-pipeline' function from Methylpy as described in *Schultz et al. (2015)*. Briefly, quality-filtering and adapter-trimming were performed using cutadapt v1.9.dev1 (*Martin, 2011*). Qualified Reads were aligned to the *E. salsugineum* v1.0 reference genome (*Yang et al., 2013*) (downloaded from: https://phytozome.jgi.doe.gov) using bowtie 2.2.4 (*Langmead and Salzberg, 2012*). Only uniquely aligned and non-clonal reads were retained. Chloroplast DNA (which is fully unmethylated) was used as a control to calculate the sodium bisulfite reaction non-conversion rate of unmodified cytosines. A binomial test was used to determine the methylation status of cytosines with a minimum coverage of three reads.

Identification of DMRs (Differential Methylated Regions) was performed using 'DMRfind' function from Methylpy pipeline as described in *Schultz et al. (2015)*. Default parameters were adopted and only DMRs with at least 5 DMSs (Differential Methylated Sites) were reported and used for subsequent analysis.

To produce metaplots, 1 kb regions upstream and downstream features of interest were divided into 20 bins each. Features of interest were also divided into total of 20 bins. Weighted methylation levels were computed as the number of methylated reads divided by the total reads for each bin.

To generate the heatmap shown in *Figure 2A*, weighted percent CHG methylation was calculated for all significant CHG DMRs with a minimum of 5 cytosines with three read coverage each in all lines (listed in *Supplementary file 2*). DMRs were then ranked by weighted %CHG methylation levels in wild type (vertical axis) and arranged by lineage (horizontal axis). To identify called DMRs reported in *Supplementary file 3*, all significant DMRs with minimum coverage requirements were filtered by cutoffs of a minimum of 10% change, relative to wild type, to be considered a hypo- or hyper- CHH or CHG DMR, and a minimum of 20% change, relative to WT, to be considered a hypo- or hyper- CG DMR. Overlaps of hyper-CG, CHG, and CHH DMRs and generation of upset plots, shown in *Figure 1—figure supplement 3*, were calculated using Intervene v.0.6.1 (*Khan and Mathelier, 2017*). To calculate the overlap of hyper-CHG DMRs between all individuals of the AtCMT3-L1 and AtCMT3-L2 lineages reported in *Figure 1—figure supplement 4*, coordinates of the called hyper-CHG DMRs reported in *Supplementary file 3* were input to the bedtools v2.27.1 fisher command (*Quinlan and Hall, 2010*). P values reported were calculated from Fisher's Exact Test with significance set at $p < 0.0004$ based on the Bonferroni correction for multiple testing.

To identify CHG-gain genes, genes were first filtered for those that had less than 1% cytosine methylation in any sequence context in wild-type. Genes were then filtered for coverage and only those with at least 10 informative CHG cytosines (min. five read coverage) in each line were assessed. Coverage corrections were separately done for: 1. wild type and all individuals of the AtCMT3-L1 and AtCMT3-L2 lineages described in *Figure 1A*; 2. wild type and all other individuals besides AtCMT3-L3T2c and the individuals of the AtCMT3-L1 and AtCMT3-L2 described in *Figure 1A*; and 3. wild type and AtCMT3-L3T2c. Genes were then called as CHG-gain genes if they showed a minimum of 5% increase in CHG methylation in *AtCMT3* expressing lines relative to wild-

type. Percent CHG methylation was calculated as the number of methylated reads mapping to CHG sites at the gene of interest divided by the total number of reads mapping to CHG sites. CHG-gain genes are listed in *Supplementary file 5*. The significance of the overlap of CHG-gain genes between individuals shown in *Figure 2—figure supplement 1* was calculated with a hypergeometric test with significance set at p<0.0004 based on the Bonferroni correction for multiple testing.

To generate the heatmap shown in *Figure 5A* and *Figure 5—figure supplement 1A*, all CHG DMRs identified in *Supplementary file 2* that overlapped the CHG-gain genes AtCMT3-L2$^{T4}$ were analyzed. Weighted methylation levels of trinucleotide sub-contexts (CNN, N = A/T/C/G) were calculated for each qualified DMR in wild type and AtCMT3-L2, T1-T6 derived from WGBS. The methylation levels of sub-contexts in wild type was used to determine their hierarchical clustering relationships. Corresponding methylation levels in ATCMT3-L2, T1-T6 were plotted in the same order.

To determine the relative number of methylated cytosines over CHG-gain genes in individuals of the AtCMT3-L2 lineage relative to AtCMT3-L2$^{T4}$ (shown in *Figure 5B*, *Figure 5—figure supplement 1B–C*), the analysis was limited to cytosines located within the regions analyzed in *Figure 5A* that had a minimum of 3 read coverage in each line and were not methylated in wild type. Methylation status was determined with a binomial test. Number of methylated cytosines in each sequence context were reported as a ratio, relative to the number of methylated cytosines in AtCMT3-L2$^{T4}$. In *Figure 5C*, the same analysis was performed as described in *Figure 5B*, except cytosines analyzed were those found in hyper-CHG DMRs defined in AtCMT3-L2$^{T4}$ that did not overlap genes and were thus annotated as repeats or intergenic regions. In *Figure 5—figure supplement 1D*, the same analysis was conducted except the regions analyzed were the hyper-CHG DMRs identified in AtCMT3-L1$^{T4}$ that overlapped AtCMT3-L1$^{T4}$ CHG-gain genes and the ratios of methylated cytosines are relative to AtCMT3-L1$^{T4}$. The same analysis was also conducted to produce *Figure 5—figure supplement 1E–F*, except the regions analyzed were the hyper CHG DMRs identified in AtCMT3-L1$^{T5}$ that overlapped AtCMT3-L1$^{T5}$ CHG gain genes (*Figure 5—figure supplement 1E*) or did not overlap genes (*Figure 5—figure supplement 1F*) and the ratios of methylated cytosines are relative to AtCMT3-L1$^{T5}$.

To identify the background levels of CG methylation shown in *Figure 5—figure supplement 2*, an equal number of genes as the number of CHG-gain genes identified in AtCMT3-L2$^{T4}$ (for *Figure 5—figure supplement 2A*) or AtCMT3-L1$^{T5}$ (for *Figure 5—figure supplement 2B*) was randomly selected from all genes that were classified as unmethylated in wild type and did not gain CHG methylation in AtCMT3-expressing lineages using the R command sample. Then an equal amount of sequence as analyzed in *Figure 5B* (for *Figure 5—figure supplement 2A*) or *Figure 5—figure supplement 1E* (for *Figure 5—figure supplement 2B*) was extracted from the randomly chosen genes, with the total number of nucleotides distributed evenly across each gene. All CG cytosines with less than three read coverage in each line assessed and cytosines found to be methylated in wild type (Shandong accession) were removed from the analysis. Coverage filtering was completed individually for the lineages assessed in each panel of *Figure 5—figure supplement 2*. Methylated CG cytosines were then identified in each lineage using a binomial test and percent CG methylation was calculated as the number of methylated CG cytosines divided by the total number of CG cytosines. This was completed for five randomly chosen sets of unmethylated genes and compared to the identical analysis completed on the CHG gain gene regions assessed in *Figure 5B* (for *Figure 5—figure supplement 2A*) or *Figure 5—figure supplement 1E* (for *Figure 5—figure supplement 2B*). Bisulfite sequencing data for *E. salsugineum* Yukon accession were from *Bewick et al. (2016)*.

## RNA sequencing mapping and analyses

Quality-filtering and adapter-trimming were performed using Trimmomatic v0.33 with default parameters (*Bolger et al., 2014*). Qualified reads were aligned to the *E. salsugineum* v1.0 reference genome using HISAT2 v2.0.5 (*Kim et al., 2015*). Gene expression (calculated as fragments per kilobase million; FPKM) values were computed using StringTie v1.3.3b (*Pertea et al., 2016*). To compare expression between wild type and *AtCMT3*-expressing lines, Log$_2$ fold change values were calculated as Log$_2$ (FPKM AtCMT3 line/FPKM wild-type). Genes with zero FPKM values were removed from expression analyses. A cutoff of ±2 Log$_2$ fold change was used to identify genes undergoing substantial changes in expression.

To conduct Gene Ontology enrichment analyses of up- and down-regulated genes, *A. thaliana* orthologs of *E. salsugineum* genes identified in *Niederhuth et al. (2016)* were utilized to extract GO annotations from TAIR (www.arabidopsis.org). These annotations were then used to identify significantly enriched GO terms with an elim Fisher's exact test (p-value<0.01) using the R package topGO (https://bioconductor.org/packages/release/bioc/html/topGO.html).

## ChIP-sequencing mapping and analyses

Quality-filtering and adapter-trimming were performed using Trimmomatic v0.33 (*Bolger et al., 2014*) with default parameters. The remaining reads were aligned to the *E. salsugineum* v1.0 reference genome using Bowtie v1.1.1 (*Langmead et al., 2009*) with the following parameters: 'bowtie -m 1 v 2 –best –strata –chunkmbs 1024 S'. Aligned reads were sorted using SAMtools v1.2 and duplicated reads were removed using SAMtools v0.1.19 (*Li et al., 2009*). ChIP-peaks were defined in wild type relative to input using HOMER 4.10 (*Heinz et al., 2010*) with the following parameters: '-region -tagThreshold 10 -size 1000 -minDist 2500 -tbp 0'. Identified peaks that were directly connected together were merged into a single region. Peaks were then further filtered by read density. The density for each merged region was defined as follows: aligned reads divided by region length. Only merged regions with density greater than 0.05 were outputted as peaks and used for subsequent analysis.

We used these ChIP-peak regions and the coordinates of CHG-gain genes to create metaplots to compare enrichment between samples. In the metaplots, mapped reads were normalized to total mapped reads for each locus of interest, and were averaged over 4 bins representing 2 kb upstream, 2 kb from the transcription start site into the gene, 2 kb from the transcription stop site into the gene, and 2 kb downstream. Finally, the average bin values were normalized to account for the number of loci.

## qRT-PCR analyses

Relative expression of the *AtCMT3* transgene to *TUB4* (Thhalv10003210m) was calculated using the double delta threshold cycle (Ct) method as $2\hat{\,}-((Average\ AtCMT3\ Ct) - (Average\ TUB4\ Ct))$ (*Livak and Schmittgen, 2001*). Average Ct values were calculated from three technical replicates.

## Characterization CHG-gain genes

Gene length and exon number for CHG-gain genes and UM genes were derived from *E. salsugineum* reference annotation files. CHG site number was calculated by scanning the whole gene sequence with a three base window and step size of 1 base. Both positive strand and negative strand were considered. The CHG frequency was calculated by normalizing CHG site number to gene length. Significance for differences in characteristics between CHG-gain and UM genes was calculated with a Wilcoxon rank-sum in the stats package of R v3.44.

Orthologs of CHG-gain genes in *A. thaliana* and their gbM status reported in *Supplementary file 9* were those identified in *Niederhuth et al. (2016)*. To determine if gbM-gain genes had more *A. thaliana* orthologs that were classified as gbM than expected by chance, a hypergeometric test was conducted using the values of 20,211 total *A. thaliana* genes with an *E. salsugineum* ortholog, with 4532 of those being classified as gbM in *A. thaliana* by *Niederhuth et al. (2016)*.

## Data availability

All sequencing data generated have been deposited into NCBI Gene Expression Omnibus under accession number: GSE128687.

## Acknowledgements

The authors would like to thank Javier Gallego-Bartolomé and Steve Jacobsen for providing the pEG302 AtCMT3 construct and Karen Schumaker for providing *E. salsugineum* seeds. RJS and FJ acknowledge support from the Technical University of Munich-Institute for Advanced Study funded by the German Excellence Initiative and the European Seventh Framework Programme under grant agreement no. 291763. RJS is supported by NSF MCB-1856143 and is a Pew Scholar in the

Biomedical Sciences, supported by The Pew Charitable Trusts. JMW is supported by an NSF NPGI Postdoctoral Fellowship (IOS-1811694).

## Additional information

### Funding

| Funder | Grant reference number | Author |
| --- | --- | --- |
| Pew Charitable Trusts | Pew Scholar in the Biomedical Sciences | Robert J Schmitz |
| National Science Foundation | NSF NPGI Postdoctoral Fellowship IOS-1811694 | Jered M Wendte |
| German Excellence Initiative and the European Seventh Framework Programme | Grant agreement no. 291763 | Frank Johannes Robert J Schmitz |
| National Science Foundation | NSF MCB-1856143 | Robert J Schmitz |

The funders had no role in study design, data collection and interpretation, or the decision to submit the work for publication.

### Author contributions

Jered M Wendte, Conceptualization, Data curation, Software, Formal analysis, Investigation, Visualization, Methodology, Writing—original draft, Writing—review and editing; Yinwen Zhang, Lexiang Ji, Data curation, Software, Formal analysis, Visualization, Methodology, Writing—review and editing; Xiuling Shi, Investigation, Methodology, Writing—review and editing; Rashmi R Hazarika, Yadollah Shahryary, Conceptualization, Software, Formal analysis, Methodology, Writing—review and editing; Frank Johannes, Conceptualization, Software, Formal analysis, Supervision, Funding acquisition, Methodology, Writing—review and editing; Robert J Schmitz, Conceptualization, Resources, Formal analysis, Supervision, Funding acquisition, Methodology, Project administration, Writing—review and editing

### Author ORCIDs

Jered M Wendte (iD) https://orcid.org/0000-0002-0663-3779
Rashmi R Hazarika (iD) http://orcid.org/0000-0002-5843-0535
Yadollah Shahryary (iD) http://orcid.org/0000-0002-9828-3373
Frank Johannes (iD) http://orcid.org/0000-0002-7962-2907
Robert J Schmitz (iD) https://orcid.org/0000-0001-7538-6663

### Decision letter and Author response

Decision letter https://doi.org/10.7554/eLife.47891.036
Author response https://doi.org/10.7554/eLife.47891.037

## Additional files

### Supplementary files

• Supplementary file 1. Sequencing statistics for all next generation sequencing data analyzed in this study.
DOI: https://doi.org/10.7554/eLife.47891.020

• Supplementary file 2. Differentially methylated regions identified by methylpy.
DOI: https://doi.org/10.7554/eLife.47891.021

• Supplementary file 3. Called hyper- and hypo-DMRs in each lineage.
DOI: https://doi.org/10.7554/eLife.47891.022

• Supplementary file 4. FPKM values for all genes determined by RNA-seq.
DOI: https://doi.org/10.7554/eLife.47891.023

• Supplementary file 5. Lists of genes that gained a minimum of 5% CHG methylation in each line.

DOI: https://doi.org/10.7554/eLife.47891.024

• Supplementary file 6. Genes with greater than (+/-) two log$_2$ fold change in expression identified in each line.

DOI: https://doi.org/10.7554/eLife.47891.025

• Supplementary file 7. P-values for Fisher's Exact tests of enrichment of CHG-gain genes in up- or down-regulated genes.

DOI: https://doi.org/10.7554/eLife.47891.026

• Supplementary file 8. Gene Ontology analysis for biologic processes enriched in up- or down-regulated genes.

DOI: https://doi.org/10.7554/eLife.47891.027

• Supplementary file 9. List of CHG-gain genes with closest *A. thaliana* ortholog and *A. thaliana* gbM status.

DOI: https://doi.org/10.7554/eLife.47891.028

• Supplementary file 10. List of *E. salsugineum* IBM1-like genes and expression status.

DOI: https://doi.org/10.7554/eLife.47891.029

• Transparent reporting form

DOI: https://doi.org/10.7554/eLife.47891.030

## Data availability

All sequencing data generated have been deposited into NCBI Gene Expression Omnibus under accession number: GSE128687.

The following dataset was generated:

| Author(s) | Year | Dataset title | Dataset URL | Database and Identifier |
|---|---|---|---|---|
| Wendte JM, Zhang Y, Ji L, Shi X, Hazarika RR, Shahrary Y, Johannes F, Schmitz RJ | 2019 | Epimutations are associated with CHROMOMETHYLASE 3-induced de novo DNA methylation | https://www.ncbi.nlm.nih.gov/geo/query/acc.cgi?acc=GSE128687 | NCBI Gene Expression Omnibus, GSE128687 |

The following previously published dataset was used:

| Author(s) | Year | Dataset title | Dataset URL | Database and Identifier |
|---|---|---|---|---|
| Bewick AJ, Ji L, Niederhuth CE, Willing EM, Hofmeister BT, Shi X, Wang L, Lu Z, Rohr NA, Hartwig B, Kiefer C, Deal RB, Schmutz J, Grimwood J, Stroud H, Jacobsen SE, Schneeberger K, Zhang X, Schmitz RJ | 2016 | On the origin and evolutionary consequences of gene body DNA methylation | https://www.ncbi.nlm.nih.gov/geo/query/acc.cgi?acc=GSE75071 | NCBI Gene Expression Omnibus, GSE75071 |

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
