## [Decision Letter]

Thank you for submitting your article "A positive feedback loop that establishes heterochromatin predisposes transcribed genes to stable epimutations" for consideration by *eLife*. Your article has been reviewed by three peer reviewers, and the evaluation has been overseen by a Reviewing Editor and Detlef Weigel as the Senior Editor. The reviewers have opted to remain anonymous.

The reviewers have discussed the reviews with one another and the Reviewing Editor has drafted this decision to help you prepare a revised submission.

Summary:

This manuscript aims at understanding the mechanism of CG gene body methylation (gbM) by expressing the Arabidopsis CMT3 gene in *Eutrema salsugineum*, a species that is lacking a CMT3 ortholog and is devoid of gbM. The authors generated several transgenic lines in *Eutrema* expressing the *Arabidopsis* CMT3 gene, and followed the establishment of CHG methylation in two independent lines over six generations. They found de novo CHG methylation on repeat sequences, intergenic sequences and on some genes that share common features with genes targeted by gbM in *Arabidopsis thaliana*. Interestingly, these genes were not marked by H3K9me2, a modification that is usually associated with CMT3 activity. Finally, it is shown that CMT3 silencing led to a fast decrease of CHG and CHH methylation and to a slow decrease of CG methylation. It is proposed that transient deposition of H3K9me2 on genes may recruit CMT3, which in turn methylates DNA in CHG context, leading to CG methylation by an unknown mechanism.

Essential revisions:

1) The main shortcoming of the manuscript is the failure to point to a mechanism through which CHG methylation mediates CG methylation. The main data to support the assumption that CHG methylation leads to gbM are those in Figure 6, showing that loss of CMT3 expression (after transgene silencing) causes a slower decline of CG methylation compared to CHG and CHH methylation. The authors made use of the fact that CMT3 became silenced in the fifth generation in the AtCMT3-L2 line to test the effect of CMT3 loss on DNA methylation. Since it is unclear if the silencing is triggered by the transgene itself or by other factors, and how quickly silencing occurred, the results in Figure 6 are not completely conclusive. Moreover, the data are based on a single line and one individual per generation.

To address this point we request the analysis of at least one other non-silenced transgenic line, as well as the analysis of transgene-free progeny instead of the apparently silenced line.

2) Plants expressing CMT3 show body CHG methylation, but not body H3K9me2. The authors seem to interpret this observation as H3K9me2 recruiting CMT3 but H3K9me2 being removed by IBM1. An alternative interpretation of the results could be that H3K9me1, rather than H3K9me2, recruits CMT3. CMT3 binds not only to H3K9me2 but also to H3K9me1, even though CMT2 does not bind to H3K9me1 (Stroud et al., 2014, Figure 2D). Therefore, it seems possible that preexisting H3K9me1 guides CMT3, but not CMT2, to introduce CHG methylation in gene bodies.

To address this point we request the comparison of H3K9me1 in transgenic and non-transgenic control plants.

3) In Figure 5A, the authors compared CHG methylation gain and expression change in each gene in the T4 plant with body CHG methylation. Their interpretation of the results is that "genic CHG methylation in AtCMT3-expressing lines is uncoupled from heterochromatin formation and transcriptional silencing, similar to gbM." A possible complication can be that transcriptome should reflect both primary effects of the body CHG methylation and indirect effects from those primary effects. The indirect effects would not correlate with CHG methylation.

Furthermore, the proportion of CHG methylated genes in downregulated genes (X<-2) seems significantly higher than those in upregulated genes (2<X) and control genes (-2<X<2). The results look consistent with transcriptome analyses in ibm1 mutants in Arabidopsis (Inagaki et al., 2017). According to that literature, genes downregulated in ibm1 mutants have significant levels of CHG methylation, while upregulated genes do not. In addition, GO analyses of upregulated genes suggest their link to immune responses. The interpretation by Inagaki et al. was that CHG methylation induces downregulation for a subset of genes, and that the upregulation of many genes reflects indirect effects, likely due to primary changes in expression of some key factor involved in immune responses.

As the results in Figure 5A seem consistent with the transcriptome of ibm1 mutants (downregulation as primary effects and upregulation as indirect effects), we suggest GO analysis of upregulated genes, and statistical test for the overrepresentation of CHG gain in downregulated genes. If GO analysis of upregulated genes show some tendency, even if not immune response, that would suggest indirect effects involved.

More generally, the question is if body CHG methylation affects gene expression. That could be clarified by examining other plants without the gain of CHG methylation, such as other transgenic lines or other generations of plants in the same line, as controls (in addition to the non-transgenic wild-type plants). Especially interesting controls might be T6 plants of the line that lost body CHG methylation while keeping body CG methylation, because that might disentangle effects of body CG and CHG methylation.

Other points:

Please address the following additional points raised by the three reviewers as much as possible.

*Reviewer #1:*

4) The manuscript suggests that an increase of CMT3 expression leads to an increase of CHG methylation (subsection “Expression of AtCMT3 in *E. salsugineum* results in increased CHG methylation”, third paragraph). It is surprising why the authors did not use the AtCMT3-L3 for their analysis, as CMT3 is higher expressed in this line than in AtCMT3-L1 (Supplementary file 4). If the model is correct, AtCMT3-L3 should exhibit a higher methylation level than lines -L1 or -L2. As the authors have generated the methylome of this line (Figure 1—figure supplement 1), they should analyze gbM in this line. In fact, it would be advisable to include also the data for the other lines, since based on Figure 1—figure supplement 1 the authors produced methylome data for all lines until the second generation.

5) Figure 4: The choice of the generations used in the analysis requires justification; why analyzing generations 1, 2, 4, 5, 8, 11 for Col-0 and generations 5, 13 for suv4/5/6 ? At least they should include the 13th Col-0 generation in the analysis, since the DNA methylation pattern can be significantly affected after few generations (Figure 1, AtCMT3-L2). It is also advisable to include additional explanations to this paragraph to allow the reader to follow what has been analyzed.

6) It is unclear how the RNA seq data were analyzed (subsection “RNA sequencing mapping and analyses”).

7) For the ChIP-seq analysis, input seems to have been retrieved (subsection “Chromatin immunoprecipitation and sequencing (ChIP-seq)”) but apparently was not used for the normalization, neither have H3 data been used to normalize (subsection “ChIP-sequencing mapping and analyses”). The analysis has to be described in more detail or repeated.

*Reviewer #2:*

8) In regard to the ways of presentation of the ChIP-seq results, they showed metaplot and browser view for H3K9me2. It might also be informative to see scatter plot, comparing H3K9me2 level between the T4 and non-transgenic plants, to see if the signal increases in a subset of genes or TE genes.

9) Based on results in Figure 4, authors discussed that gain of CG methylation rate was significantly lower in Arabidopsis suvh4/5/6 mutant than in WT, but loss of CG methylation was not significantly different. The background statistics was not clear to me.

*Reviewer #3:*

10) The manuscript often refers to "targeting" of CMT3 to genes, which to me implies an active process. But the authors reach the conclusion (Discussion, last paragraph) that gbm is likely a passive byproduct of having a functional CMT3 enzyme. It's not that CMT3 is specifically targeted to genes, but that CMT3 acts in non-heterchromatic regions with some low frequency. It is suggested that the authors reconsider their use of the word targeting.

11) One of the results that most strongly supports the authors' model is that CG methylation is retained (and CHG and CHH methylation is lost) in CHGhyper genes after CMT3 is silenced in line L2 (Figure 6A and 6B). This led me to wonder why *Eutrema* lacks all gbm – presumably the loss of CMT3 occurred relatively recently in its evolutionarily history since its closest relatives retain CMT3 and gbm. Shouldn't some CG methylation still be present? Can the authors date the loss of CMT3 in *Eutrema* and does the total absence of gbm in this species fit with the timing of that loss and what we know about rates of mCG gain and loss in gene bodies over time?

12) Based on analysis of repeat methylation, the authors suggest that AtCMT3 preferentially targets heterochromatin over genes. There are alternative interpretations for these data. DNA methylation was reduced in all contexts in L2T6 in heterochromatin (Figure 6C), although it remained higher than in genes (Figure 6B). But the remaining methylation could be due to other maintenance and de novo pathways also being more active in those regions (MET1, RdDM, CMT2), rather than any residual CMT3 activity preferentially being directed to heterochromatin.

13) The portion of the manuscript about epimutations rates in suvh4/5/6 in *Arabidopsis* was a distraction from the main message. The effects on CG methylation gain, while statistically significant, do not appear particularly strong. I recommend removing this section from the paper.

14) The right panel of Figure 1—figure supplement 5 does not support the conclusion that higher levels of CMT3 expression are correlated with increased global CHG methylation. The R2 value of 0.59 is driven by two points and many samples have high CHG methylation but relatively low CMT3 expression. This doesn't seem like a key conclusion of the paper, and the authors should be more cautious in their interpretation.

---

## [Author Response]

Essential revisions:1) The main shortcoming of the manuscript is the failure to point to a mechanism through which CHG methylation mediates CG methylation. The main data to support the assumption that CHG methylation leads to gbM are those in Figure 6, showing that loss of CMT3 expression (after transgene silencing) causes a slower decline of CG methylation compared to CHG and CHH methylation. The authors made use of the fact that CMT3 became silenced in the fifth generation in the AtCMT3-L2 line to test the effect of CMT3 loss on DNA methylation. Since it is unclear if the silencing is triggered by the transgene itself or by other factors, and how quickly silencing occurred, the results in Figure 6 are not completely conclusive. Moreover, the data are based on a single line and one individual per generation.To address this point we request the analysis of at least one other non-silenced transgenic line, as well as the analysis of transgene-free progeny instead of the apparently silenced line.

These are excellent suggestions and we have added the new data as the reviewers have requested. We conducted the same analysis we completed for AtCMT3-L2 in AtCMT3-L1, which did not demonstrate silencing of the transgene. In this case, the relative methylation levels over genes in all contexts (CG, CHG, and CHH) remained similar over generational time. This result is included in new Figure 5—figure supplement 1D.

We also completed an alternative approach where we crossed the AtCMT3-L1T5 transgene expressing line to wild type (non-transgenic) to segregate out the transgene. We analyzed two F2 progeny where the transgene was segregated out and one F2 progeny that still contained the transgene. In the F2 progeny that did not encode the transgene, the ectopic genic CHG and CHH methylation that was present in the in the transgenic parent was lost but the genic CG methylation was maintained, similar to the line where the transgene was silenced. In contrast, the F2 progeny that still encoded the transgene maintained similar levels of genic methylation in all contexts as the transgenic parent. This result is included in the new Figure 5—figure supplement 1E. We also verified that the levels of newly methylated genic CG sites in these lines, as well as the lines where AtCMT3 was silenced (AtCMT3-L2T5-T6), were higher than background (due to bisulfite non-conversion or epimutations that occur independently of CMT3), by comparing control sets of un-methylated genes that did not gain CHG methylation and by comparing to an additional, non-transgenic accession of *E. salsugineum* (Yukon). These controls are included in new Figure 5—figure supplement 2.

These new results are described in the text in subsection “Ectopic genic CG methylation is preferentially maintained following loss of AtCMT3 expression”, fourth paragraph.

2) Plants expressing CMT3 show body CHG methylation, but not body H3K9me2. The authors seem to interpret this observation as H3K9me2 recruiting CMT3 but H3K9me2 being removed by IBM1. An alternative interpretation of the results could be that H3K9me1, rather than H3K9me2, recruits CMT3. CMT3 binds not only to H3K9me2 but also to H3K9me1, even though CMT2 does not bind to H3K9me1 (Stroud et al., 2014, Figure 2D). Therefore, it seems possible that preexisting H3K9me1 guides CMT3, but not CMT2, to introduce CHG methylation in gene bodies.To address this point we request the comparison of H3K9me1 in transgenic and non-transgenic control plants.

This is an intriguing hypothesis, however, we conducted H3K9me1 ChIP-seq in transgenic and non-transgenic lines and found that, similar to H3K9me2, there does not appear to be an enrichment of H3K9me1 in genes that gain CHG methylation before or after the introduction of the transgene (see new supplementary Figure 3—figure supplement 2A-C).

These new results are discussed in the text in the last paragraph of the subsection “CHG methylation in gene bodies is not associated with stable H3K9 methylation”.

We do agree with the reviewer that the possibility of an initial presence of H3K9me1, even if transient, could help explain the phylogenetic correlation between encoding CMT3, but not CMT2, and the presence gene body methylation across plant species. Once CMT3 methylates CWG cytosines, H3K9 methylation may then be stabilized as H3K9me2, which could be bound by CMT2, promoting CWA methylation. We have included this possibility in our Discussion and hypothetical model. See new model Figure 6 and the Discussion section.

3) In Figure 5A, the authors compared CHG methylation gain and expression change in each gene in the T4 plant with body CHG methylation. Their interpretation of the results is that "genic CHG methylation in AtCMT3-expressing lines is uncoupled from heterochromatin formation and transcriptional silencing, similar to gbM." A possible complication can be that transcriptome should reflect both primary effects of the body CHG methylation and indirect effects from those primary effects. The indirect effects would not correlate with CHG methylation.Furthermore, the proportion of CHG methylated genes in downregulated genes (X<-2) seems significantly higher than those in upregulated genes (2<X) and control genes (-2<X<2). The results look consistent with transcriptome analyses in ibm1 mutants in Arabidopsis (Inagaki et al., 2017). According to that literature, genes downregulated in ibm1 mutants have significant levels of CHG methylation, while upregulated genes do not. In addition, GO analyses of upregulated genes suggest their link to immune responses. The interpretation by Inagaki et al. was that CHG methylation induces downregulation for a subset of genes, and that the upregulation of many genes reflects indirect effects, likely due to primary changes in expression of some key factor involved in immune responses.As the results in Figure 5A seem consistent with the transcriptome of ibm1 mutants (downregulation as primary effects and upregulation as indirect effects), we suggest GO analysis of upregulated genes, and statistical test for the overrepresentation of CHG gain in downregulated genes. If GO analysis of upregulated genes show some tendency, even if not immune response, that would suggest indirect effects involved.More generally, the question is if body CHG methylation affects gene expression. That could be clarified by examining other plants without the gain of CHG methylation, such as other transgenic lines or other generations of plants in the same line, as controls (in addition to the non-transgenic wild-type plants). Especially interesting controls might be T6 plants of the line that lost body CHG methylation while keeping body CG methylation, because that might disentangle effects of body CG and CHG methylation.

We conducted RNA sequencing of nine of the *AtCMT3* expressing plants we analyzed, including the T6 plant that demonstrated loss of *AtCMT3* expression, and in each case, we found most of the CHG-gain genes showed little to no change in expression (Figure 4A, Figure 4—figure supplement 1). As suggested by the reviewer, in each of these lines we conducted a statistical test to determine if the CHG gain genes are enriched in either down- or up- regulated genes identified genome-wide, using a cutoff of a +/- 2 log_2_ fold change. We found no enrichment in either down- or up- regulated genes (see new Supplementary file 7).

To assess the possibility of indirect effects of AtCMT3 expression, we conducted a GO enrichment analysis of up and down regulated genes in each line. We found that in each line, both up and down regulated genes showed significant enrichments in various abiotic stress response GO terms, but did not see an overrepresentation of immune response genes (see new Supplementary file 8). Despite this general trend for abiotic stress response genes, we found that there was no GO term consistently identified across all lineages, which lead us to conclude that it is unlikely that these changes in expression are related to the transgene expression either directly or indirectly. These results are discussed in the text in the subsection “CHG methylation in gene bodies is not associated with transcriptional silencing”.

With these additional results, we believe our original conclusion that "genic CHG methylation in AtCMT3-expressing lines is uncoupled from heterochromatin formation and transcriptional silencing, similar to gbM," is still valid. An important distinction of the genic CHG methylation we identified in AtCMT3-expressing *E. salsugineum* and that found in *A. thaliana ibm1* mutants is that we did not detect stable H3K9 methylation associated with genic CHG methylation in the *Eutrema* lineages. Our prediction is that *Eutrema* IBM1 is actively removing H3K9 methylation, which prevents DNA methylation from leading to heterochromatin formation and silencing (See model Figure 6). This contrasts with *A. thaliana ibm1* mutants, where both heterochromatin signals (DNA and histone methylation) are established over genes, which affects transcription.

Other points:Please address the following additional points raised by the three reviewers as much as possible.Reviewer #1:4) The manuscript suggests that an increase of CMT3 expression leads to an increase of CHG methylation (subsection “Expression of AtCMT3 in E. salsugineum results in increased CHG methylation”, third paragraph). It is surprising why the authors did not use the AtCMT3-L3 for their analysis, as CMT3 is higher expressed in this line than in AtCMT3-L1 (Supplementary file 4). If the model is correct, AtCMT3-L3 should exhibit a higher methylation level than lines -L1 or -L2. As the authors have generated the methylome of this line (Figure 1—figure supplement 1), they should analyze gbM in this line. In fact, it would be advisable to include also the data for the other lines, since based on Figure 1—figure supplement 1 the authors produced methylome data for all lines until the second generation.

Our initial inclination was to include all lines in the main figures. However, simply due to space constraints, including all the data resulted in crowded figures that were difficult to read. We wanted to avoid confusion in our discussions of the dynamics of DNA methylation over generational time and chose to focus the main figures on the two lineages that we had samples over the full course of six generations while limiting the analyses of additional lineages to the supplementary figures.

Our analyses of the additional lineages included in the supplementary material confirmed the results presented in the main text figures and we agree with the reviewer that this should be stated more emphatically. We have therefore discussed these results more explicitly in the main text to emphasize that the two L3 individuals with the highest CMT3 expression also had the highest levels of CHG methylation and number of genes gaining CHG methylation:

“The results demonstrated a significant correlation between the levels of AtCMT3 expression and genome-wide CHG methylation levels (R^2^ = 0.8828, p = 1.669 x 10^-4^, Figure 1—figure supplement 5, see Supplementary file 4 for FPKM values). This was especially notable for two T2 generation plants of AtCMT3-L3, which had the highest AtCMT3 expression levels (233.7 and 372.5 FPKM for AtCMT3-L3T2 and T2b, respectively) and the highest genome-wide percent CHG methylation (27% and 32%, for T2a and T2b, respectively).”

and:

“Indeed, based on RNA-seq assessment of AtCMT3 expression, the levels of AtCMT3 expression and number CHG-gain genes were correlated (R^2^ = 0.7951, p = 0.001) (Figure 2—figure supplement 2, Supplementary file 4). Again, two T2 generation individuals of the AtCMT3-L3 lineage, which had the highest AtCMT3 expression (Supplementary file 4), also had the highest number of CHG-gain genes (5,566 and 6,346 CHG-gain genes for AtCMT3-L3T2 and T2b, respectively) (Supplementary file 5).”

5) Figure 4: The choice of the generations used in the analysis requires justification; why analyzing generations 1, 2, 4, 5, 8, 11 for Col-0 and generations 5, 13 for suv4/5/6 ? At least they should include the 13th Col-0 generation in the analysis, since the DNA methylation pattern can be significantly affected after few generations (Figure 1, AtCMT3-L2). It is also advisable to include additional explanations to this paragraph to allow the reader to follow what has been analyzed.

At the request of reviewer #3, we have removed this analysis so as not to distract from the main conclusions of the paper.

6) It is unclear how the RNA seq data were analyzed (subsection “RNA sequencing mapping and analyses”).

We have expanded this section of the Materials and methods to make this clearer:

“Quality-filtering and adapter-trimming were performed using Trimmomatic v0.33 with default parameters (Bolger, Lohse, and Usadel, 2014). […] Genes with zero FPKM values were removed from expression analyses. A cutoff of +/- 2 Log_2_ fold change was used to identify genes undergoing substantial changes in expression.”

7) For the ChIP-seq analysis, input seems to have been retrieved (subsection “Chromatin immunoprecipitation and sequencing (ChIP-seq)”) but apparently was not used for the normalization, neither have H3 data been used to normalize (subsection “ChIP-sequencing mapping and analyses”). The analysis has to be described in more detail or repeated.

We added in more detail to the Materials and methods section to clarify how we analyzed the ChIP-seq data:

“Quality-filtering and adapter-trimming were performed using Trimmomatic v0.33 (Bolger et al., 2014) with default parameters. […] Finally, the average bin values were normalized to account for the number of loci.”

Reviewer #2:8) In regard to the ways of presentation of the ChIP-seq results, they showed metaplot and browser view for H3K9me2. It might also be informative to see scatter plot, comparing H3K9me2 level between the T4 and non-transgenic plants, to see if the signal increases in a subset of genes or TE genes.

Since our ChIP experiments were conducted at time points separated by months, due to the multi-generational nature of the study, we focused our analyses on major trends in enrichment over certain genomic features of interest (heterochromatin and genes that gained CHG methylation in transgenic lines). We appreciate there are certainly many ways to display these data, but ultimately decided that metaplots and representative browser views were the best way to accurately portray the general trends while remaining conservative in our approach.

9) Based on results in Figure 4, authors discussed that gain of CG methylation rate was significantly lower in Arabidopsis suvh4/5/6 mutant than in WT, but loss of CG methylation was not significantly different. The background statistics was not clear to me.

At the request of reviewer #3, we have removed this analysis so as not to distract from the main conclusions of the paper.

Reviewer #3:10) The manuscript often refers to "targeting" of CMT3 to genes, which to me implies an active process. But the authors reach the conclusion (Discussion, last paragraph) that gbm is likely a passive byproduct of having a functional CMT3 enzyme. It's not that CMT3 is specifically targeted to genes, but that CMT3 acts in non-heterchromatic regions with some low frequency. It is suggested that the authors reconsider their use of the word targeting.

We agree this terminology can be confusing and we have removed “targeting” from our references to CMT3-localization to genes.

11) One of the results that most strongly supports the authors' model is that CG methylation is retained (and CHG and CHH methylation is lost) in CHGhyper genes after CMT3 is silenced in line L2 (Figure 6A and 6B). This led me to wonder why Eutrema lacks all gbm – presumably the loss of CMT3 occurred relatively recently in its evolutionarily history since its closest relatives retain CMT3 and gbm. Shouldn't some CG methylation still be present? Can the authors date the loss of CMT3 in Eutrema and does the total absence of gbm in this species fit with the timing of that loss and what we know about rates of mCG gain and loss in gene bodies over time?

This is an interesting question; however, we would require additional phylogenetic sampling to estimate when in evolutionary history the ancestors of modern *Eutrema salsugineum* lost CMT3.

12) Based on analysis of repeat methylation, the authors suggest that AtCMT3 preferentially targets heterochromatin over genes. There are alternative interpretations for these data. DNA methylation was reduced in all contexts in L2T6 in heterochromatin (Figure 6C), although it remained higher than in genes (Figure 6B). But the remaining methylation could be due to other maintenance and de novo pathways also being more active in those regions (MET1, RdDM, CMT2), rather than any residual CMT3 activity preferentially being directed to heterochromatin.

This is a good point and we have conducted an additional experiment that has allowed us to test this more thoroughly. We crossed the AtCMT3-L1T5 line to wild type in order to segregate out progeny that no longer encode the transgene to better evaluate maintenance of transgene induced ectopic methylation following transgene removal. We found that, similar to the L2T6 lineage, transgene induced methylation was maintained at higher levels in heterochromatin than it was in genes in the CHG and CHH contexts in progeny with the transgene crossed out. This supports the reviewer’s hypothesis that other pathways can preferentially maintain methylation in these regions. However, we also note that the relative CHG methylation in particular was reduced by a further ~50% when the transgene was completely removed compared to the L2T6 lineage which supports the possibility that the residual CMT3 activity remaining in this line was preferentially directed to heterochromatic regions.

We have included these results in Figure 5—figure supplement 1F and discussed them in the last paragraph of the subsection “AtCMT3 preferentially methylates heterochromatin relative to genes”.

13) The portion of the manuscript about epimutations rates in suvh4/5/6 in Arabidopsis was a distraction from the main message. The effects on CG methylation gain, while statistically significant, do not appear particularly strong. I recommend removing this section from the paper.

We have removed this section from the paper to avoid distraction and updated the text and figures accordingly. We have also opted to change the title since we are now focusing the paper mainly on the role of CMT3 in the initiation of gbM. Our new title is: “Epimutations are associated with CHROMOMETHYLASE 3-induced de novo DNA methylation”.

14) The right panel of Figure 1—figure supplement 5 does not support the conclusion that higher levels of CMT3 expression are correlated with increased global CHG methylation. The R2 value of 0.59 is driven by two points and many samples have high CHG methylation but relatively low CMT3 expression. This doesn't seem like a key conclusion of the paper, and the authors should be more cautious in their interpretation.

We agree that the correlation of *AtCMT3* expression and genome-wide CHG methylation was not as strong when measured by qRT-PCR compared to RNA-seq (0.59 vs. 0.88 R2, respectively). However, taken together, we think the results do support the likelihood that variation in *AtCMT3* expression is a contributing factor to overall CHG methylation levels, which has been noted previously in *A. thaliana* mutant backgrounds that alter *CMT3* expression (e.g. Cell. 2014 Jul 3; 158(1): 98–109). We have altered the text to present this conclusion more cautiously:

“We also examined the relationship between genome-wide CHG methylation and AtCMT3 expression using qRT-PCR, including plants from additional lineages, and found a weaker although significant relationship (R^2^ = 0.5932, p = 0.025, Figure 1—figure supplement 1 and Figure 1—figure supplement 5). Taken together, AtCMT3 expression is likely one factor contributing to genome-wide CHG levels in these lines.”